# Fast, multiplexable and efficient somatic gene deletions in adult mouse skeletal muscle fibers using AAV-CRISPR/Cas9

Marco Thürkauf [1], Shuo Lin[1], Filippo Oliveri[1], Dirk Grimm [2,3,4], Randall J. Platt [5,6] & Markus A. Rüegg [1] ✉

Molecular screens comparing different disease states to identify candidate genes rely on the availability of fast, reliable and multiplexable systems to interrogate genes of interest. CRISPR/Cas9-based reverse genetics is a promising method to eventually achieve this. However, such methods are sorely lacking for multi-nucleated muscle fibers, since highly efficient nuclei editing is a requisite to robustly inactive candidate genes. Here, we couple Cre-mediated skeletal muscle fiber-specific Cas9 expression with myotropic adeno-associated virus-mediated sgRNA delivery to establish a system for highly effective somatic gene deletions in mice. Using well-characterized genes, we show that local or systemic inactivation of these genes copy the phenotype of traditional gene-knockout mouse models. Thus, this proof-of-principle study establishes a method to unravel the function of individual genes or entire signaling pathways in adult skeletal muscle fibers without the cumbersome requirement of generating knockout mice.

With the advent of -omics technologies that allow to correlate molecular signatures with specific disease states of cells or tissues, there is an increasing need for methods to interrogate the function of genes and pathways. Traditionally, forward and reverse genetics using targeted mutagenesis in combination with transgenesis has been used. More recently, clustered regularly interspaced short palindromic repeats (CRISPR)-mediated genome editing has become the method of choice for gene engineering in many species and tissues[1].

When it comes to skeletal muscle tissue, studying gene function in vivo is particularly challenging. Skeletal muscle is one of the largest organs constituting up to 50% of the mammalian body mass[2]. The size and the fact that muscle fibers, which are the functional contractile units of skeletal muscle, form a syncytium with hundreds of myonuclei in a common cytosol, represent a substantial challenge for somatic gene inactivation. Therefore, the method of choice for functional gene interrogation studies in muscle remains transgenic mice generated via the Cre-loxP system. However, generation of transgenic mice requires

extensive breeding, making functional interrogation of multiple genes cumbersome and time consuming.

Effective methods for somatic gene perturbation would offer huge advantages for screening multiple muscle gene candidates. While RNA interference, which can silence a target gene by introducing short hairpin (sh) RNAs[3], can acutely silence gene expression in muscle fibers[4,5], prolonged elimination of a gene product requires sustained, high expression of the shRNA. The introduction of viruses, in particular adeno-associated viruses (AAV), as vehicles for delivering shRNAs, opened the possibility of systemic administration[6]. However, due to the lack of tissue-specific control of shRNA expression, gene silencing usually occurs in all transduced cells. While next-generation AAV capsids with designed tropism towards skeletal muscle tissue[7–9] may improve the off-tissue targeting, all of them also target myocytes in the heart. Another challenge for somatic gene targeting of muscle fibers is the overall heterogeneity of the tissue. Almost half of the nuclei in skeletal muscle derive from non-fiber cells, such as muscle stem cells

[1]Biozentrum, University of Basel, Basel, Switzerland. [2]Department of Infectious Diseases/Virology, Section Viral Vector Technologies, Medical Faculty, Heidelberg University, Heidelberg, Germany. [3]BioQuant, University of Heidelberg, Heidelberg, Germany. [4]German Center for Infection Research (DZIF) and German Center for Cardiovascular Research (DZHK), Heidelberg, Germany. [5]Department of Biosystems Science and Engineering (D-BSSE), ETH Zurich, Basel, Switzerland. [6]Department of Chemistry, University of Basel, Basel, Switzerland. ✉e-mail: markus-a.ruegg@unibas.ch

(MuSC), endothelial cells, fibro-adipogenic precursors (FAPs), Schwann cells or tenocytes[10] and perturbation of their function often affects muscle fibers as well. Therefore, for rapid functional gene interrogation in skeletal muscle fibers, an efficient, multiplexable, and muscle fiber-specific gene editing approach is sorely needed.

Here we establish a versatile tool for local and systemic skeletal muscle fiber-specific gene knockout. This tool couples the advantages of CRISPR with recently developed, highly efficacious, AAV9-derived viral capsids by using (i) mice engineered to constitutively or inducibly express Cas9 in skeletal muscle fibers and (ii) delivering single guide (sg) RNAs with the myotropic AAVMYO[8]. By targeting key genes, we demonstrate that this system is capable of potently altering signaling pathways, destroying neuromuscular junctions, and stimulating muscle hypertrophy without needing to generate germline gene-of-interest deletions.

## Results

### Constitutive expression of Cas9 in skeletal muscle fibers
To express Cas9 at high levels in skeletal muscle fibers, we crossed Cre-dependent Rosa26$^{Cas9-EGFP}$ knockin mice[11] with mice that either express Cre recombinase constitutively in skeletal muscle fibers[12] (scheme in Fig. 1a) or after tamoxifen injection[13] (scheme in Fig. S2a, b). The resulting Cre-positive Rosa26$^{Cas9-EGFP}$ knockin mice were called Cas9 muscle knock-in (Cas9mKI) and inducible Cas9mKI (iCas9mKI) mice, respectively. Expression of Cas9 in Cas9mKI mice was confirmed by immunohistochemistry for GFP (Fig. 1b). By Western blot analysis, high levels of Cas9 were detected in all muscles tested (Fig. 1c). In liver, Cas9 was not detected while traces were eventually detected in heart (Fig. 1c). Body mass (Fig. 1d), muscle mass (Fig. 1e), overall fiber size distribution (Fig. 1c) and fiber-type composition and distribution (Fig. S2a–c) were indistinguishable between control and Cas9mKI mice. Likewise, all the measured functional parameters, such as grip strength and ex vivo force, were the same in control and Cas9mKI mice (Fig. 1g–j; Fig. S2e–g). In addition, neuromuscular junctions (NMJs) looked identical in control and Cas9mKI mice when visualized in whole-mount preparations (Fig. S1d). Finally, *tibialis anterior* (TA) muscle response to cardiotoxin-induced injury was the same in Cas9mKI and control mice (Fig. S1h, i). In the iCas9mKI mice, Cas9 was highly expressed in adult skeletal muscles 14 days after tamoxifen injection but not detected in heart or liver (Fig. S2c–e). Together, these data show that Cas9 is highly expressed in skeletal muscle fibers of Cas9mKI and iCas9mKI mice and they indicate that this high expression does not affect muscle physiology.

### Robust in vivo gene editing using local AAV9-mediated sgRNA delivery into Cas9mKI mice
To test whether high Cas9 expression would allow gene perturbation in skeletal muscle to an extent required to lower protein levels, we selected *Prkca*, which codes for protein kinase Cα (PKCα). We selected PKCα based on a combination of our experience characterizing PKCα as an mTORC2 target in the brain[14], the availability of antibodies for Western blot analyses and because CRISPR has been used to successfully eliminate PKCα in the retina[15]. As low PKCα levels, due to loss of mTORC2, do not affect skeletal muscle[16], we could determine the effectiveness of the system independent of secondary effects by the loss of PKCα. Beside the published sgRNA (called sgPKCα-1), we tested an additional sgRNA (sgPKCα-2) and included a non-targeting sgRNA (sgNT). Cultured C2C12 myoblasts were transfected with a plasmid encoding the U6 promoter-driven sgRNA followed by an EFS promoter-driven Cas9, the P2A self-cleavage peptide and puromycin N-acetyltransferase, which confers puromycin resistance to transfected cells (Fig. S3a). After puromycin selection, C2C12 myoblasts were differentiated into myotubes for five days. All selected cells expressed Cas9 and those co-expressing sgPKCα-1 or sgPKCα-2, but not sgNT, showed strongly reduced levels of PKCα (Fig. S3b, c). As

sgPKCα-1 and sgPKCα-2 showed similar efficiency, we selected the published sgPKCα-1[15] for further characterization. To quantify the number of insertions and deletions of bases (indels), we sequenced genomic DNA in the region targeted by sgPKCα-1 and used the method of "Tracking of Indels by Decomposition" (TIDE). The total DNA-editing efficiency for sgPKCα-1 was 62.4 ± 1.6% (Fig. S3d), with the majority of deletions lacking 1 base pair (−1 bp) followed by insertions of +1 bp (Fig. S3e).

Based on the high efficiency of sgPKCα-1 in editing *Prkca* and lowering the amount of PKCα in cultured C2C12 cells, we next tested the overall genome editing efficiency by injecting AAV9 that expressed either one (AAV9-1sgPKCα-1) or three copies (AAV9-3sgPKCα-1); as a control we used AAV9 expressing a non-targeting sgRNA (AAV9-sgNT). To monitor transduction efficiency, the constructs also included a CMV-driven coding sequence for tdTomato (see scheme in Fig. S4a). The same amount of the different AAVs (3 × 10$^{11}$ vg) was injected into *tibialis anterior* (TA) muscle of adult Cas9mKI mice together with 5 mU neuraminidase, an enzyme that has been shown to increase AAV9 transduction efficiency in skeletal muscle[17,18]. Six weeks after injection, mice were analyzed. Relative mass of the virus-injected TA or EDL muscle was not altered (Fig. S4b), indicating that the procedure was well tolerated. Editing of the *Prkca* locus, measured by TIDE, was significantly increased from 15.6 ± 3.3% to 26.2 ± 2.4% when using the construct with 3 copies (Fig. S4c), indicating that the amount of the expressed sgRNA contributes to the genome editing efficiency.

Because of the superior performance of AAV9-3sgPKCα-1, all subsequent experiments used this construct (schematic presentation in Fig. 2a). Transduction efficiency using AAV9 + neuraminidase treatment was high and homogenous in TA muscle, as monitored by tdTomato staining in cross sections (Fig. 2b) and reached 99 ± 9.6 AAV genomes/nucleus (Fig. 2c). The probable virus leakage into the blood stream resulted in strong liver (117.8 ± 20.0 AAV genomes/nucleus) and significant heart (19 ± 3.3 AAV genomes/nucleus) transduction (Fig. 2c). To determine genome editing efficiency in the different tissues, we used TIDE for the sgPKCα-1-targeted *Prkca* locus. The background editing signal in AAV9-sgNT-transduced TA muscle was 1.4 ± 0.5%, while the AAV9-3sgPKCα-1-injected muscle reached 20.3 ± 1.0% editing (Fig. 2d). In heart and liver, no significant *Prkca* editing was detected, indicating that the combination of intramuscular AAV9 injection and HSA-driven Cas9 expression allows for very specific gene editing in skeletal muscle fibers. As a consequence of CRISPR/Cas9-mediated DNA editing in TA muscle, PKCα protein was strongly diminished in AAV9-3sgPKCα-1-injected compared to AAV9-sgNT-injected muscle (Fig. 2e, f). The low amount of PKCα still detected in AAV9-3sgPKCα-1-transduced TA muscle may also derive from other muscle-resident cells that express *Prkca* transcripts[10]. Together, these data show that AAV9-mediated sgRNA delivery in combination with neuraminidase treatment markedly reduces PKCα protein specifically in the targeted skeletal muscle of Cas9mKI mice.

### Improved editing efficiency with AAVMYO for local sgRNA delivery into Cas9mKI mice
In a next step, we tested a peptide-displaying AAV9 capsid variant (called AAVMYO) with superior skeletal muscle fiber tropism[8], for the delivery of sgRNA as this variant may allow high transduction efficiency in local and systemic injection without the need of neuraminidase. First, we compared the efficiency of AAVMYO-3sgPKCα-1 with AAV9-3sgPKCα-1 (without neuraminidase treatment) by injecting 3 × 10$^{11}$ vg of each, or a PBS control into TA muscle (Fig. 3a). Six weeks post-injection, tdTomato expression was visibly higher in TA muscle as well as the nearby *extensor digitorum longus* (EDL) and *gastrocnemius* (GAS) muscles of AAVMYO-injected than AAV9-injected muscles (Fig. 3b). Cross sections from TA (Fig. 3b), EDL and GAS muscles (Fig. S5a) as well as Western blot quantification from TA muscle

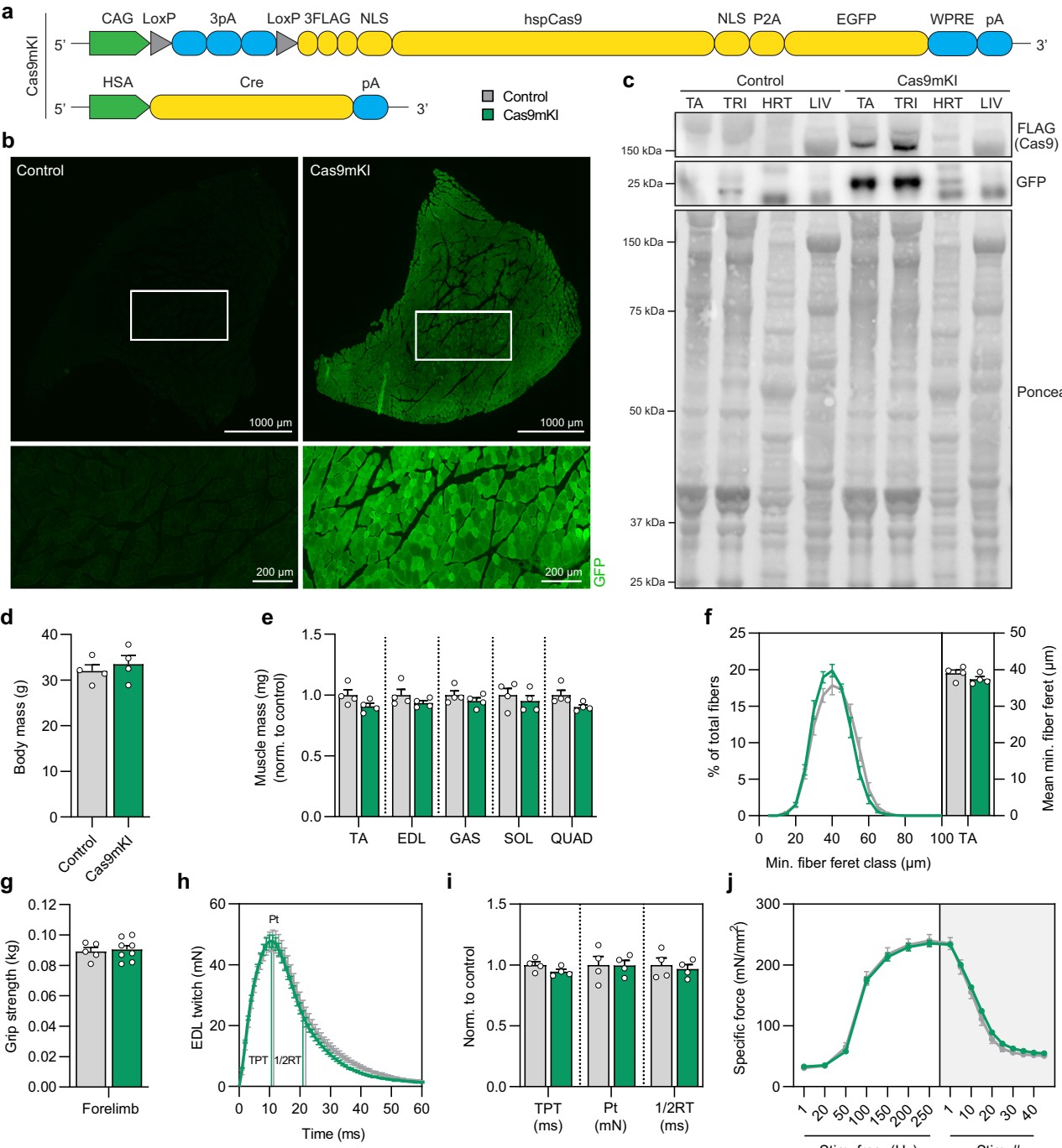

**Fig. 1 | Validation of Cas9mKI mice. a** Schematic of the Cas9mKI mouse model. CAG: cytomegalovirus (CMV) enhancer fused to the chicken beta-actin promoter; LoxP: locus of X-over P1; pA: polyadenylation signal; FLAG: FLAG-tag; NLS: nuclear localization signal; hspCas9: humanized *Streptococcus pyogenes* Cas9; P2A: 2A self-cleaving peptide; EGFP: enhanced green fluorescent protein; WPRE: woodchuck hepatitis virus post-transcriptional regulatory element; HSA: human α-skeletal actin; Cre: Cre recombinase. **b** Cross sections of *tibialis anterior* (TA) muscle stained for EGFP (green) in control and Cas9mKI mice. **c** Western blot analysis of lysates from TA, *triceps brachii* (TRI), heart (HRT), and liver (LIV) of control and Cas9mKI mice using antibodies against the FLAG-tag or GFP. Only TA and TRI muscles of Cas9mKI mice are positive for the FLAG-tag and GFP; no expression was detected in HRT or LIV. Ponceau staining was included as loading control. **d** Body mass of 19-week-old control and Cas9mKI mice. **e** Relative mass of TA, *extensor digitorum longus* (EDL), *soleus* (SOL), *gastrocnemius* (GAS), and *quadriceps* (QUAD) muscles

from control and Cas9mKI mice. **f** Minimal fiber feret distribution (left) and mean minimal fiber feret (right) of muscle fibers from TA of control and Cas9mKI mice. **g** For limb grip strength of control and Cas9mKI mice. **h** Ex vivo twitch response of isolated EDL muscle from Cas9mKI and control mice. Peak twitch (Pt), time-to-peak twitch (TPT) and half-relaxation time (1/2RT) are indicated. **i** Quantification of ex vivo twitch response parameters (TPT, Pt, 1/2RT) of isolated EDL muscle from Cas9mKI and control mice. **j** Force-frequency curve (left) and fatigue response to multiple stimulations (right) of EDL muscle from control and Cas9mKI mice. Data are means ± SEM. For (**b**–**f**) and (**h**–**j**), n = 4 mice. For **g**, n = 5 (control) and 8 (Cas9mKI) mice. None of the data are significantly different between control and Cas9mKI mice (P > 0.05) using unpaired student's two-sided t-test. Experimental scheme in (**a**) was adapted from Platt et al.[11]. Source data and precise p-values are provided as a source data file.

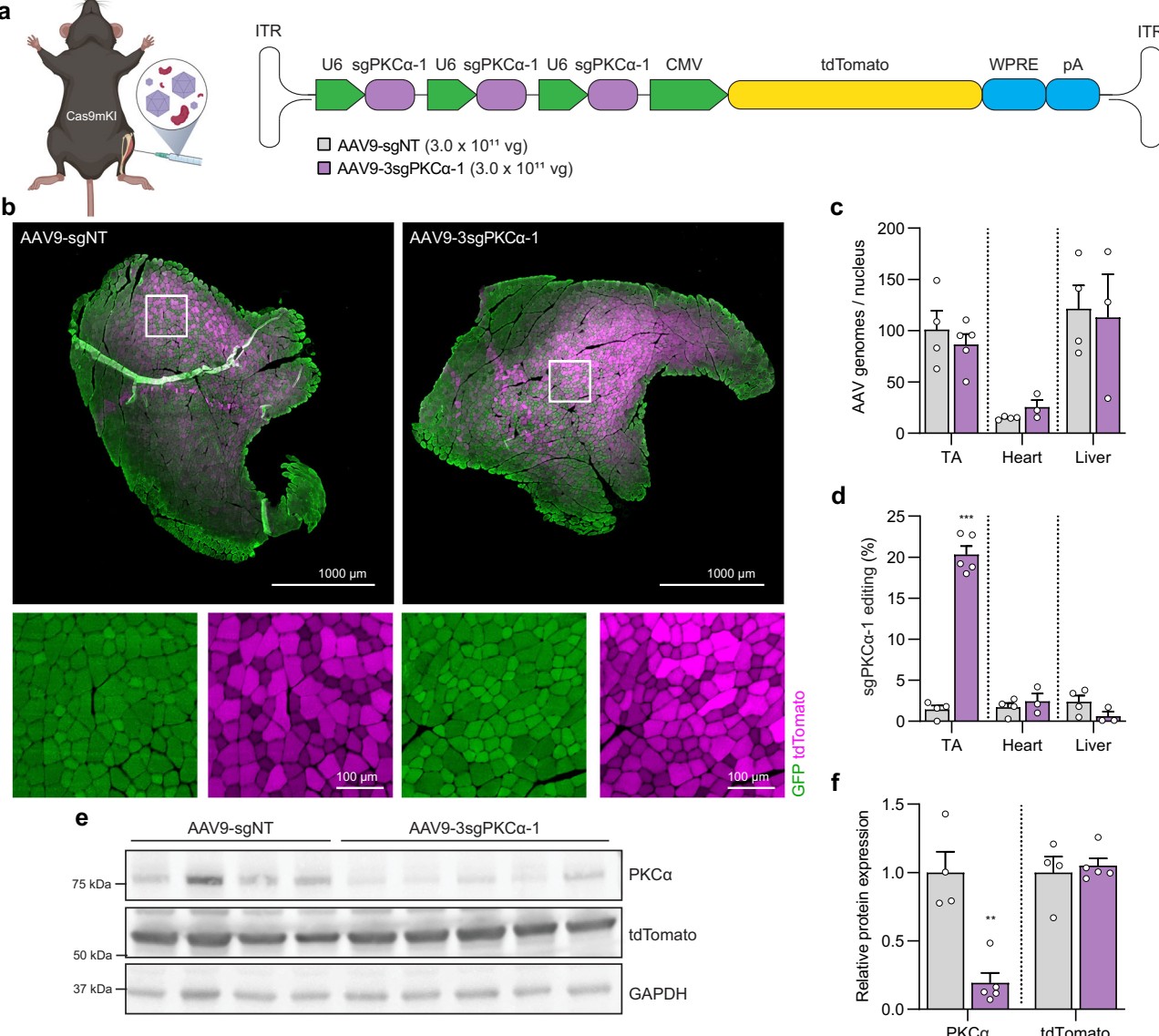

**Fig. 2 | Loss of PKCα in TA muscle upon AAV9/neuraminidase-mediated delivery of sgRNA to Cas9mKI mice. a** Schematic illustration of experimental procedure and targeting construct. For abbreviations, see legend to Fig. 1. **b** Cross section of *tibialis anterior* (TA) muscle stained for Cas9/GFP (green), tdTomato (magenta), and DAPI (blue), 6 weeks after injection of neuraminidase plus AAV9-sgNT or AAV9-3sgPKCα-1. **c** Quantification of AAV genomes per nucleus in TA muscle, heart, and liver in AAV9-sgNT (gray) and AAV9-3sgPKCα-1-injected (purple) mice. **d** Total INDEL formation analysis by TIDE on sgPKCα-1 target locus of TA muscle, heart, and liver injected with AAV9-sgNT and AAV9-3sgPKCα-1. **e** Western blot analysis and **f** quantification of PKCα and tdTomato expression in TA muscle 6 weeks post-injection with AAV9-sgNT (gray) or AAV9-3sgPKCα-1 (purple). Data are means ± SEM. For (**b**–**d**) and (**f**), $n = 4$ (AAV9-sgNT: TA, heart, liver), 5 (AAV9-3sgPKCα-1: TA), and 3 (AAV9-3sgPKCα-1: heart, liver) mice. Statistical analysis used unpaired student's two-sided t-test. *$P < 0.05$, **$P < 0.01$, ***$P < 0.001$. Experimental scheme in (**a**) was created with BioRender.com. Source data and precise *p*-values are provided as a source data file.

(Fig. 3c, d) further confirmed higher tdTomato expression in AAVMYO- than AAV9-injected muscles. Average transduction by AAVMYO, judged by AAV genomes/nucleus, was at least 2.5 times higher than by AAV9 for all muscles, including the heart, while transduction of the liver was markedly lower (Fig. S5b). The superior transduction efficiency of AAVMYO over AAV9 upon intramuscular injection is in line with previous observations upon systemic administration of AAVMYO and AAV9[8]. As a consequence of the more efficient transduction, the amount of PKCα was also more strongly diminished in AAVMYO-3sgPKCα-1-transduced TA (Fig. 3c, d), EDL (Fig. S5c, d) and GAS (Fig. S5f, g) compared to AAV9-3sgPKCα-1. TIDE analysis showed a higher total editing efficiency with AAVMYO-3sgPKCα-1 (22.4 ± 1.4%) than with AAV9-3sgPKCα-1 (17.2 ± 0.9%) in TA muscle (Fig. 3e). Compared to the intramuscular injection of AAVMYO into TA muscle, measures of transduction efficiency (PKCα knock down; tdTomato

amount; TIDE analysis) remained rather high in the adjacent EDL and GAS muscles, while they dropped with AAV9 (Fig. S5c–h).

To more precisely map genome editing frequencies in the genomic DNA surrounding the sgPKCα-1 target site, we performed next-generation sequencing (NGS) of TA muscle DNA (Fig. 3f and Supplementary Data 1). The sum of all observed mutations with NGS was comparable to TIDE analysis; with average mutations of 18.7 ± 0.6% for AAV9 and 23.1 ± 1.3% for AAVMYO (Fig. 3g). Independent of the AAV capsid variant, the most frequent indels were short deletions (Fig. 3h). To test whether introduction of sgPKCα-1 caused off-target editing, we also sequenced the genome in the top four off-target sites as predicted by the CRISPR-design tool CRISPOR[19]. No significant sequence alterations were detected at these loci (Fig. 3i).

Denervation and hence loss of muscle contraction has an immediate effect on gene expression in myonuclei and results after a

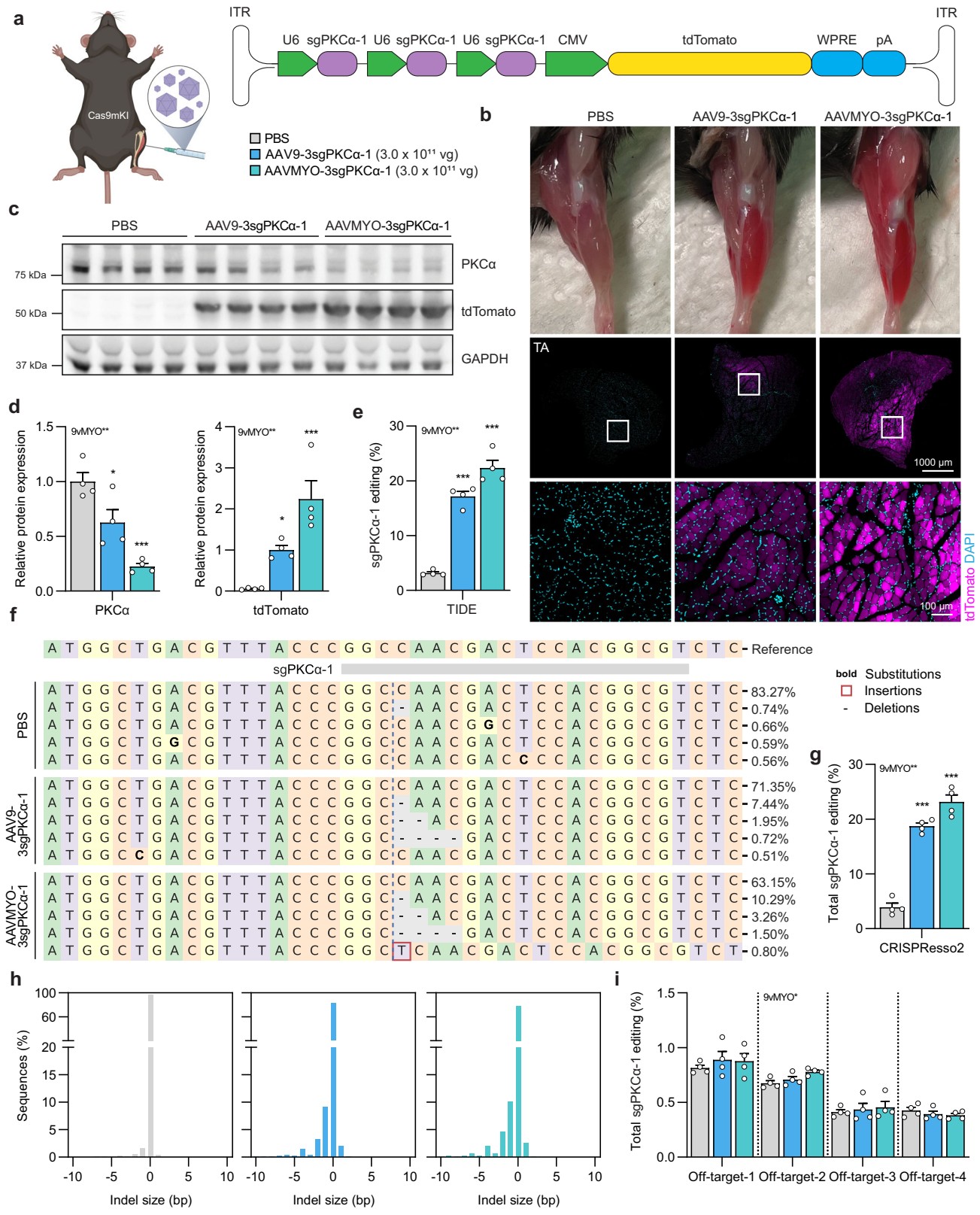

few days in exuberant muscle atrophy. It indirectly also affects satellite cells and many other muscle-resident mononuclear cells. As somatic gene deletion may affect innervation, we wanted to assess whether the gene editing system would also work during acute denervation. To test this, we injected AAVMYO-3sgPKCα-1 ($3 \times 10^{11}$ vg) or PBS (as a control) into TA muscle of Cas9mKI mice before unilateral sciatic nerve transection 6 weeks later and then analyzed muscle 14 days later (Fig. S6a).

The denervation-induced loss of muscle mass was not different between PBS and AAVMYO-3sgPKCα-1-injected mice (Fig. S6b). Importantly, denervation did not affect the overall PKCα knockdown efficiency or expression of the denervation marker HDAC4 (Fig. S6c, d). There was a slight decrease in the total percentage of genome editing by sgPKCα-1 (Fig. S6e), which is likely due to the increase in non-muscle fiber cells following denervation that do not express

**Fig. 3 | AAVMYO-mediated sgPKCα-1 delivery into TA muscle results in strong reduction of PKCα. a** Schematic presentation of the experimental procedure. For abbreviations, see legend to Fig. 2. **b** Representative images of the injected hindlimb and of cross sections of *tibialis anterior* (TA) muscle stained for tdTomato (magenta) and DAPI (blue), 6 weeks post-intramuscular injection of PBS or AAVs ($3.0 \times 10^{11}$ vg) into Cas9mKI mice. **c** Western blot analysis and quantification **d** for PKCα and tdTomato in TA muscle lysates of Cas9mKI mice injected with PBS (gray), AAV9-3sgPKCα-1 (light blue), or AAVMYO-3sgPKCα-1 (cyan) 6 weeks post-injection. **e** Total INDEL formation analysis by TIDE on sgPKCα-1 target locus. **f** Representative sequence frequency table of reads using DNA isolated from TA under the different conditions covering the sgPKCα-1 target region. **g** Relative number of modified reads under the different conditions in the sgPKCα-1 target region. **h** INDEL size histogram indicating mutation distribution at the sgPKCα-1 target region in TA muscle of Cas9mKI mice. Conditions are injection of PBS (light gray, left), AAV9-3sgPKCα-1 (light blue, middle), or AAVMYO-3sgPKCα-1 (cyan, right). **i** Total amount of mutated reads of amplicons covering the top four predicted off-target loci in the different experimental paradigms. There is no difference in the modified reads compared to PBS injection. Data are means ± SEM. $n = 4$ mice. Statistical significance is based on one-way ANOVA with Fishers LSD post-hoc test. *$P < 0.05$, **$P < 0.01$, ***$P < 0.001$. Experimental scheme in (**a**) was created with BioRender.com. Source data and precise *p*-values are provided as a source data file.

Cas9[20]. Together, our data show that AAVMYO-mediated sgRNA delivery into TA muscle induces robust and specific in vivo gene perturbation.

## AAVMYO supersedes AAV9 for systemic sgRNA delivery

To evaluate efficiency for systemic gene editing, we next injected $1 \times 10^{14}$ vg/kg of AAV9-3sgPKCα-1 or AAVMYO-3sgPKCα-1 into the tail vein of 6-week-old Cas9mKI mice and collected tissues 6 weeks later (scheme Fig. 4a). Expression of tdTomato was visually higher at autopsy and strikingly higher in cross sections of multiple muscles in mice injected with AAVMYO than with AAV9 (Fig. 4b). Similar results were obtained for the heart (Fig. S8a). Numbers of viral genomes per nucleus were 4- to 6-fold higher in limb muscles (TA, EDL, SOL and TRI) and more than 13-fold higher in the diaphragm (DIA) with AAVMYO than AAV9 (Fig. 4c). In line with the high transduction efficiency, AAVMYO-3sgPKCα-1 induced 2.2- to 7.6-fold higher *Prkca* editing rates across different muscles than AAV9-3sgPKCα-1 (Fig. 4d). The most striking difference was seen in DIA muscle, where AAVMYO-3sgPKCα-1 induced 19.1 ± 0.9% DNA editing while AAV9-3sgPKCα-1 induced only 2.5 ± 0.9%. Western blot analysis for tdTomato and PKCα confirmed the superior systemic transduction of muscle tissue by AAVMYO-3sgPKCα-1, with higher tdTomato expression and stronger reduction in PKCα protein abundance than with AAV9-3sgPKCα-1 (Fig. 4e–g).

We also tested systemic administration of additional capsid variants of AAVMYO, called AAVMYO2 and AAVMYO3, that were originally selected for their liver de-targeting qualities[7], which can be an advantage for clinical applications. AAVMYO2 and AAVMYO3 were less efficient than AAVMYO, but superior to AAV9, at transducing skeletal muscle and therefore eliciting gene editing events (Fig. S7). Liver transduction was 519.0 ± 51.3 AAV genomes/nucleus with AAV9 and dropped to 79.7 ± 18.4 AAV genomes/nucleus with AAVMYO, whereas AAVMYO2 (2.3 ± 0.5 AAV genomes/nucleus) and AAVMYO3 (1.9 ± 0.4 AAV genomes/nucleus) were largely excluded from liver (Fig. S8a, b). Heart transduction by AAVMYO was superior to AAV9 and to AAVMYO2 and AAVMYO3 (Fig. S8a, b). To investigate whether the *Prkca* locus was edited in heart or liver, we conducted the TIDE analysis. Compared to PBS injection, ~5% of the *Prkca* locus in the heart DNA was edited while no significant editing was detected in liver, irrespective of the AAV variant (Fig. S8c). These results further support the muscle specificity of this AAV-CRISPR/Cas9 system.

## AAVMYO-CRISPR/Cas9-mediated knockdown recapitulates conditional knockout model phenotypes for MuSK and myostatin/activin signaling

To further test the system, we asked whether we could recapitulate muscle phenotypes of knockout mice by targeting genes known to play a fundamental role in the regulation of muscle structure and growth. We first chose to target the receptor tyrosine kinase MuSK, the signaling component of the Lrp4/MuSK receptor complex for motor neuron-released agrin[21]. MuSK is essential for the formation and maintenance of the NMJ[4,22,23] and auto-antibodies against MuSK can cause myasthenia gravis[24], a disease leading to NMJ loss. As AAVMYO transduces skeletal muscle fibers with high efficiency, we omitted

tdTomato and instead focused on maximizing *Musk* gene editing by inserted seven different sgRNAs into the constructs directed against exons localized in the 5' region of the *Musk* gene (targeting sites see: Fig. S10a). In a first set of experiments, we injected AAVMYO-7sgMusk ($1.5 \times 10^{13}$ vg/kg) or PBS (as a control) into the lateral tail vein of Cas9mKI (scheme Fig. 5a). By following the body weight, we noted that AAVMYO-7sgMusk-injected Cas9mKI started to lose weight after 14 days and reached 20% loss, which is the humane endpoint for euthanization, 20 days post injection (Fig. 5b). Their all- and forelimb grip strength was significantly lower than in controls (Fig. 5c) and they developed a severe kyphosis indicative of muscle weakness (Fig. 5d). Mice also showed signs of muscle fibrillation and ataxia, suggestive of denervation. To test this hypothesis, we measured mass in bulbar, fore- and hindlimb muscles. Indeed, all muscles of AAVMYO-7sgMusk-injected Cas9mKI were severely atrophic compared to controls (Fig. 5e). CRISPR/Cas9 editing resulted in the reduction of *Musk* mRNA by at least 90% in all muscles examined (Fig. 5f). As a consequence, MuSK could not be detected at NMJs in EDL muscle and the loss of MuSK resulted in the very strong reduction of acetylcholine receptor (AChR) clusters (Fig. 5g). The presynaptic motor nerve terminals, visualized by a mixture of the SV2 (directed against synaptic vesicle glycoprotein 2A) and the 2H3 antibody (directed against the neurofilament-M protein) were still innervating the muscle fibers (Fig. 5g). This loss of postsynaptic structures upon MuSK depletion is consistent with the results of transgenic mice deficient for *Musk*[22,23]. Thus, our data show that the AAVMYO-CRISPR/Cas9 system generates a somatic gene knockout whose phenotype is identical to germline-based methods.

Next, we tested whether this system would also allow to restrict the depletion of MuSK to one or a few muscles. This might be advantageous as loss of genes that are essential for muscle function (such as MuSK) will result in respiratory failure and death of the mice. As AAVMYO transduces the diaphragm well (see Fig. 4), respiratory failure may jeopardize the in-depth analysis of limb muscles. To test for local perturbation, we injected different doses of AAVMYO-7sgMusk into the right TA muscle of adult Cas9mKI mice (scheme Fig. 6a) and monitored the mice for 5 weeks. As controls, we chose to inject either AAVMYO-7sgMusk into wild-type (i.e., not expressing Cas9 in muscle) or injected PBS into Cas9mKI mice. As the two control conditions did not differ, we pooled data for further analysis. As expected, we detected a dose-dependent increase in the number of AAV genomes/nucleus (Fig. 6b) and dose-dependent decrease in *Musk* mRNA expression in the injected TA muscle (Fig. 6c). However, despite intramuscular administration, mice injected with the highest dose of $3 \times 10^{11}$ vg lost body mass after 14 days and reached 20% loss after 21 days (Fig. 6d). Mice injected with second highest dose ($1 \times 10^{11}$ vg) needed to be euthanized by 35 days, while no body mass loss was observed with the two lowest doses (Fig. 6d). Analysis of hindlimb muscles in the injected leg showed a dose-dependent decline in mass, which became significant compared to controls starting at a dose of $3.3 \times 10^{10}$ vg (Fig. 6e). Examination of the NMJs in EDL muscle, identified by the presence of the presynaptic nerve terminals, showed that MuSK and AChRs were largely lost from the postsynapse in AAVMYO-

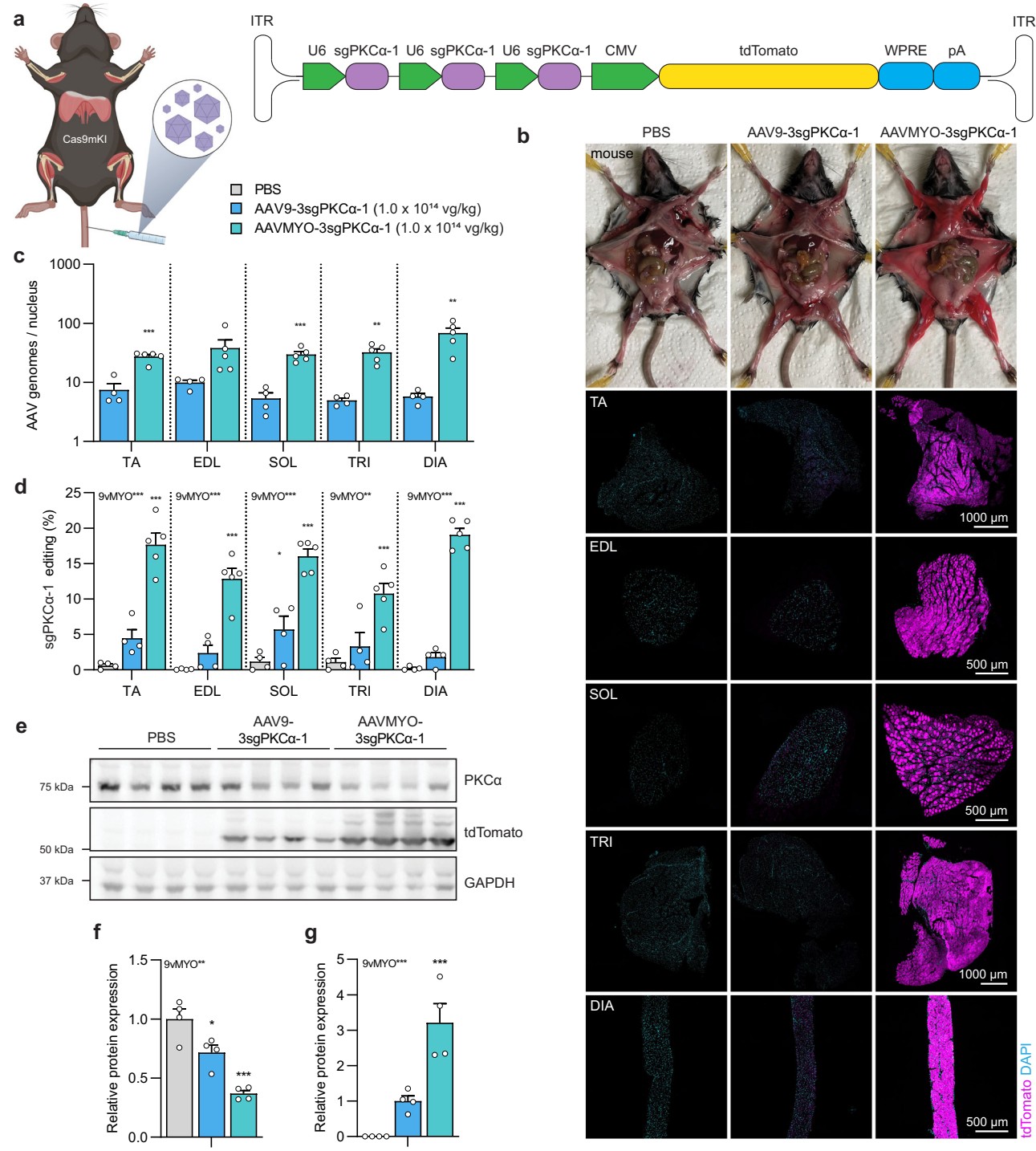

**Fig. 4 | Systemic administration of AAVMYO-3sgPKCα-1 *via* the tail vein into Cas9mKI mice reduces PKCα protein. a** Schematic presentation of the experimental procedure. For abbreviations, see legend to Fig. 2. **b** Representative images of dissected mice and cross sections of *tibialis anterior* (TA), *extensor digitorum longus* (EDL), *soleus* (SOL), *triceps brachii* (TRI), or diaphragm (DIA) muscle stained for tdTomato (magenta) and DAPI (blue), 6 weeks post-intravenous injection of PBS or AAV (1.0 × 10¹⁴ vg/kg) into Cas9mKI mice. **c** Distribution of AAV in TA, EDL, SOL, TRI, and DIA upon intravenous injection of AAV9-3sgPKCα-1 (light blue) and AAVMYO-3sgPKCα-1 (cyan) into Cas9mKI mice. **d** Total INDEL formation analysis by

TIDE on sgPKCα-1 target locus. **e** Western blot analysis and quantification (**f, g**) for PKCα (**f**) and tdTomato (**g**) in TA muscle of Cas9mKI mice injected with PBS (gray), AAV9-3sgPKCα-1 (light blue) or AAVMYO-3sgPKCα-1 (cyan). Data are means ± SEM. For (**b**–**d**), *n* = 4 (control), 4 (AAV9), and 5 (AAVMYO) mice. For (**f**) and (**g**), *n* = 4 mice. Significance was determined using one-way ANOVA with Fishers LSD post-hoc test (**d, f, g**) or unpaired student's two-sided t-test (**c**). *$P < 0.05$, **$P < 0.01$, ***$P < 0.001$. Experimental scheme in (**a**) was created with BioRender.com. Source data and precise *p*-values are provided as a source data file.

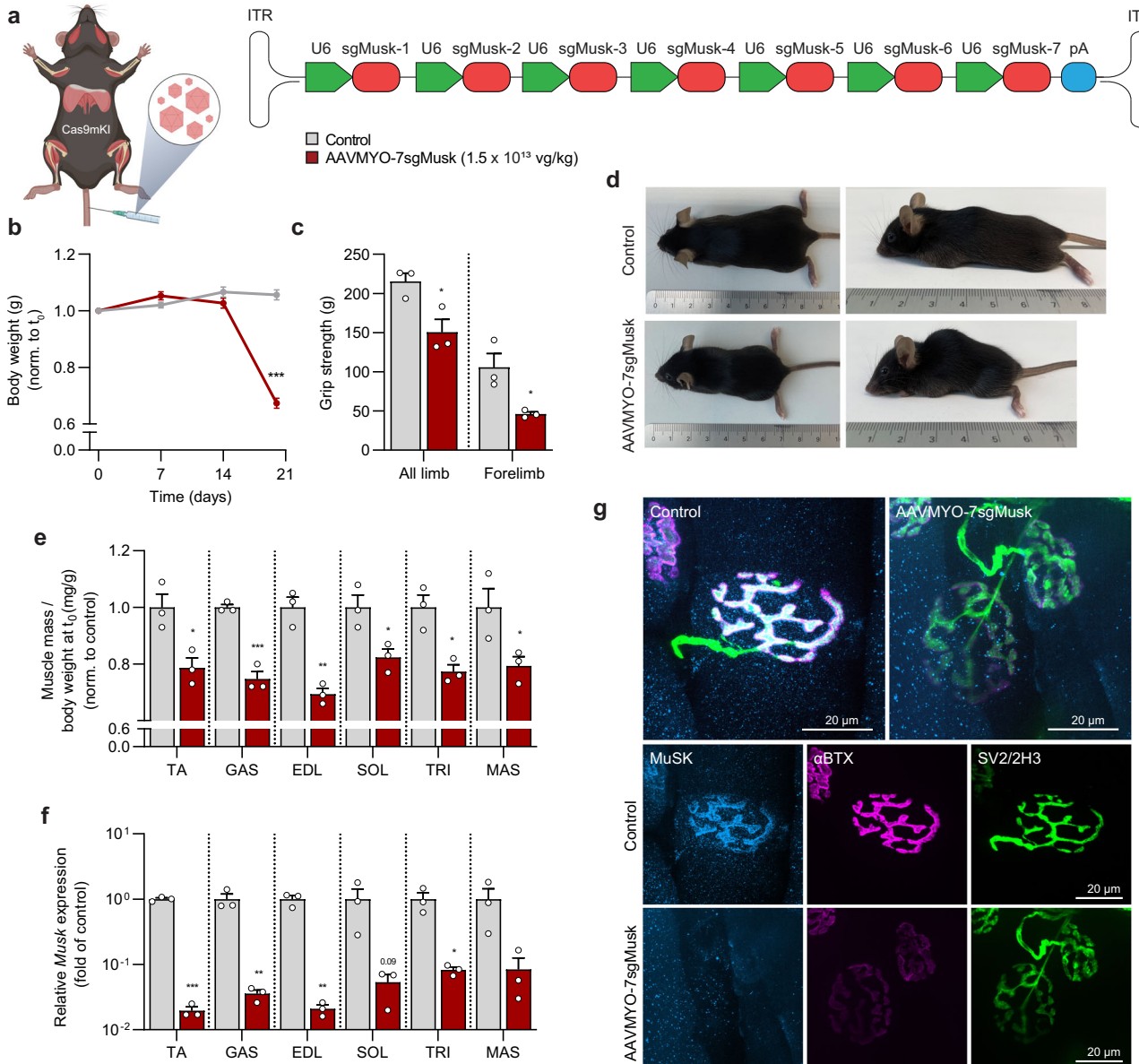

**Fig. 5 | AAVMYO-CRISPR/Cas9 mediates systemic knockout of Musk and results in the loss of NMJs. a** Schematic presentation of the experimental procedure. For abbreviations, see legend to Fig. 2. **b** Body weight progression of controls (gray) and AAVMYO-7sgMusk-injected Cas9mKI mice (red). **c** All-limb and forelimb grip strength of control and AAVMYO-7sgMusk-injected Cas9mKI mice, 20 days post injection. **d** Representative photograph of control and AAVMYO-7sgMusk-injected Cas9mKI mice, 20 days post-injection. **e** Changes in mass of *tibialis anterior* (TA), *gastrocnemius* (GAS), *extensor digitorum longus* (EDL), *soleus* (SOL), *triceps brachii* (TRI), and *masseter* (MAS) muscle of AAVMYO-7sgMusk-injected Cas9mKI mice, compared to controls. **f** Relative mRNA expression of Musk in TA, GAS, EDL, SOL, TRI, and MAS muscle of AAVMYO-7sgMusk-injected Cas9mKI mice.

**g** Representative images of whole-mount preparations of EDL muscles of controls and Cas9mKI mice injected with AAVMYO-7sgMusk. The presynaptic nerve terminals are stained with a mixture of antibodies directed against synaptic vesicle glycoprotein 2A (SV2; green) and neurofilament (2H3; green). Fluorescently-labeled α-bungarotoxin (αBTX; magenta) was used to visualize postsynaptic AChRs. MuSK protein was stained using a specific antibody (cyan). Data are means ± SEM. $n = 3$ mice. Statistical significance is based on unpaired student's two-sided t-test comparing to control. $^*P < 0.05$, $^{**}P < 0.01$, $^{***}P < 0.001$. Experimental scheme in (**a**) was created with BioRender.com. Source data and precise *p*-values are provided as a source data file.

7sgMusk-injected Cas9mKI mice, irrespective of the dose (Fig. 6f; Fig. S9a). Quantification of the staining intensity for MuSK and AChR in GAS muscle showed that both proteins were strongly reduced in AAVMYO-7sgMusk-injected Cas9mKI mice compared to controls (Fig. 6g, h). Loss of AChRs at the NMJ impairs synaptic transmission and leads to functional muscle denervation, which in turn, causes re-expression of several synaptic genes along the entire muscle fiber[20,25]. To quantify the extent of denervation, we measured expression of mRNA coding for AChRα (*Chrna1*), the embryonic AChRγ subunit (*Chrng*) and growth arrest and DNA damage-inducible 45a (*Gadd45a*). The abundance of all transcripts was more than 10-times higher in all

AAVMYO-7sgMusk-injected TA muscles of Cas9mKI mice than in controls (Fig. 6i–k).

As shown above, Cas9mKI mice injected with the highest dose of AAVMYO-7sgMusk ($3 \times 10^{11}$ vg) rapidly lost body weight and needed to be euthanized after 21 days, indicative of systemic perturbation. To test this idea, we examined muscles of the contralateral, non-injected leg. At the time of euthanization (21 and 35 days post injection), the two highest doses ($3 \times 10^{11}$ and $1 \times 10^{11}$ vg) caused a significant muscle mass loss while the two low doses did not affect this parameter (Fig. S9b). The number of AAV genomes/nucleus was between 0.5 and 0.7 in the three low doses and at $3.8 \pm 1.5$ at the highest dose (Fig. S9c).

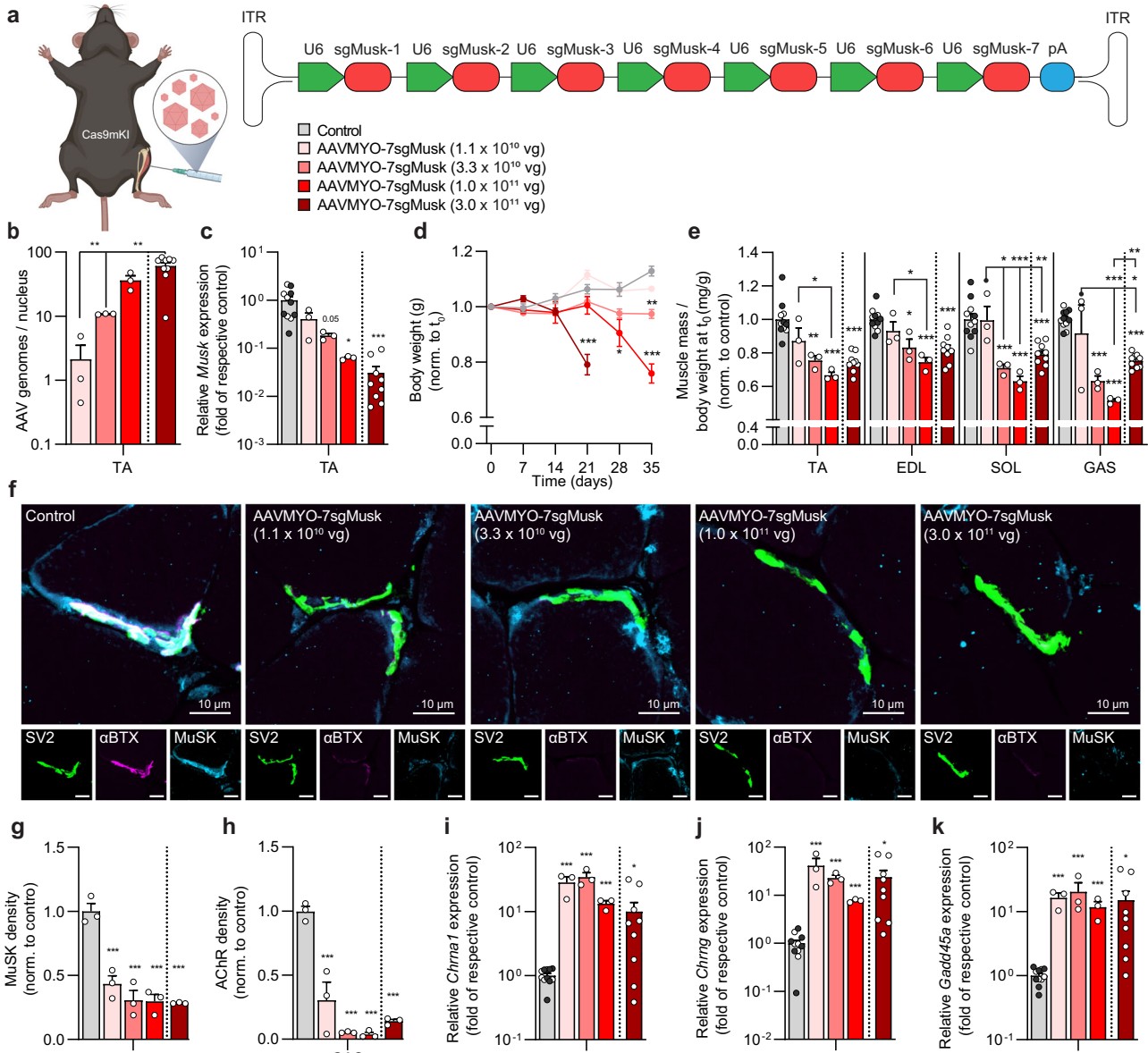

**Fig. 6 | AAVMYO-CRISPR/Cas9 mediates local knockout of *Musk* and results in the loss of NMJs. a** Schematic presentation of the experimental procedure using different amounts of sgRNA-delivering AAVMYO. For abbreviations, see legend to Fig. 2. **b** Distribution of AAV in AAVMYO-7sgMusk-injected (red colors) *tibialis anterior* (TA) muscle of Cas9mKI mice at the indicated doses. **c** Relative mRNA expression of *Musk* in AAVMYO-7sgMusk-injected TA muscle of Cas9mKI mice or controls, that included PBS-injected Cas9mKI mice (white dots) and AAVMYO-7sgMusk-injected wild-type mice (black dots). Values in the two control groups did not differ and were therefore combined. **d** Body weight progression of controls and AAVMYO-7sgMusk-injected Cas9mKI mice at the indicated doses. **e** Changes in muscle mass of TA, *extensor digitorum longus* (EDL), *soleus* (SOL), and *gastrocnemius* (GAS) AAVMYO-7sgMusk-injected Cas9mKI mice, compared to controls. **f** Representative cross section images of GAS muscles of controls and Cas9mKI mice injected with the indicated amount of AAVMYO-7sgMusk. The presynaptic nerve terminals and MuSK are stained with antibodies directed against SV2 (green) or MuSK (cyan), respectively. Fluorescently-labeled α-bungarotoxin (αBTX; magenta) was used to visualize postsynaptic AChRs. Image quantification of MuSK (**g**) and AChR (**h**) mean density, normalized to NMJ area. **i–k** Relative mRNA expression of denervation marker genes as indicated in AAVMYO-7sgMusk-injected TA muscles of Cas9mKI mice and controls. Note that Cas9mKI mice injected with the highest AAVMYO-7sgMusk dose were analyzed at 3 weeks post-injection while all the other mice were analyzed 5 weeks post-injection. Data are means ± SEM. For (**b–e**) and (**i–k**), $n = 11$ (Control (5 PBS, 6 AAVMYO)), 3 ($1.1 \times 10^{10}$ vg, $3.3 \times 10^{10}$ vg, $1.0 \times 10^{11}$ vg), and 9 ($3.0 \times 10^{11}$ vg) mice. For (**f–h**), $n = 3$ mice. Statistical significance is based on one-way ANOVA with Tukey's post-hoc test (**b–e**, **g**, **h**) and unpaired student's two-sided t-test comparing to control (**i–k**). *$P < 0.05$, **$P < 0.01$, ***$P < 0.001$. Experimental scheme in (**a**) was created with BioRender.com. Source data and precise *p*-values are provided as a source data file.

The highest dose also resulted in a significantly lower expression of *Musk* in the contralateral TA muscle and the diaphragm (Fig. S9d, e). These results indicate that the supposedly low amount of AAVMYO circulating in the blood after intramuscular injection seemed sufficient to perturb gene function in remotely-positioned skeletal muscles. One of the reasons for this high efficiency could be the use of 7 different sgRNAs to target *Musk*. The get an estimate for this, we analyzed DNA-editing for each sgRNA at the *Musk* locus (see scheme in Fig. S10a)

using amplicon-NGS and CRISPResso2[26]. The mean editing efficiency at the target site for each sgMusk upon injection of the highest AAVMYO-7sgMusk dose of $3 \times 10^{11}$ vg, was between 10.5% and 30.0% (Fig. S10b; Supplementary Data 1). With the exception of sgMusk-6, none of the sgRNAs showed significant editing at the CRISPOR[19]-predicted primary off-target site (Fig. S10c; Supplementary Data 1). As we also suspected that the multiple sgRNAs could result in large deletions in the *Musk* locus, we PCR-amplified *Musk* cDNA using primers flacking the region

targeted by the 7 sgRNAs (see scheme in Fig S10a). In control mice, four fragments of 642, 754, 785, and 815 bp-length were amplified (representing isoforms from alternative mRNA splicing[27]). In contrast, the PCR product of AAVMYO-7sgMusk-injected Cas9mKI mice appeared as a smear, indicative of multiple deletions (Fig. S11a). Interestingly, the appearance of the smear was strongest in muscle injected with a highest dose of AAVMYO-7sgMusk and concomitant with the lowest level of *Musk* transcripts determined by RT-qPCR (Fig. 6c). To detect deletions within the 7 sgMusk loci, PCR products derived from *Musk*-cDNA were sequenced and mapped to the genome. In control mice, read junctions (gray) of the PCR products always mapped to individual exons and included the three annotated and the one non-annotated splice variants (dotted gray lines) of *Musk* (Fig. S11b). However, upon CRISPR/Cas9-mediated editing of *Musk*, many additional read junctions (red lines) were observed that did not map those in control mice (Fig. S11b). These data are evidence that the use of 7 different sgRNAs for *Musk* results in a high number of indels at each sgRNA-target site and that they cause large genomic deletions between different sgRNAs. We therefore conclude that the use of multiple sgRNAs in combination with the high expression of Cas9 in all skeletal muscle fibers make the AAVMYO-CRISPR/Cas9 system highly efficient in deleting the gene of interest.

As dosing of AAVMYO to restrict its action to the injected muscle might be a challenge for different targets, we reasoned that local injection of AAV9 together with neuraminidase could eventually be an alternative. To test this idea, we injected AAV9-7sgMusk at an intermediate ($3.3 \times 10^{10}$ vg) and a high ($3.0 \times 10^{11}$ vg) dose (scheme Fig. S12a). Injected Cas9mKI mice did not lose any body weight irrespective of the dose (Fig. S12b). Analysis of hind limb muscle mass revealed a dose-dependent reduction in mass in the injected leg (Fig. S12c), but not in the muscles of the contralateral leg (Fig. S12d). Similarly, a dose-dependent decline in *Musk* mRNA was detected in the injected TA muscle (Fig. S12e) and AChRs largely disappeared from the NMJ in the EDL muscle (Fig. S12f), indicative of the loss of a functional NMJ. This caused a strong denervation response indicated by the increase of *Chrna1*, *Chrng*, and *Gadd45a* transcripts (Fig. S12g–i). Together, these results show that AAV9-7sgMusk/neuraminidase injection might be an alternative to restrict gene perturbation to the injected muscle without the challenge to eventually cause systemic effects.

To test whether our system could also drive gain of muscle function, we next targeted myostatin (GDF-8), a TGF-β family protein secreted by skeletal muscle that acts as an inhibitor of muscle size[28]. Deletion of *Mstn* in mice results in robust muscle hypertrophy[29] and naturally occurring *Mstn* null-mutants cause hypermuscularity in many species, including cows and humans[30,31]. Myostatin signals through a combination of type-2 and type-1 receptors. This signaling pathway is also activated by several other ligands, including activin. The two ligand-binding receptors are activin A receptor type-2/IIA (ACVR2A or ACTRIIA) and type-2/IIB (AAVR2B or ACTRIIB). The activin A type-2 receptors are partially redundant as targeting both receptors elicits stronger muscle hypertrophy than deletion of each receptor individually[32]. Upon ligand-binding, the type-2 receptors form a complex with type-1 activin A receptor-like kinase-4 (ALK4) and ALK5, which are also partially redundant, to trigger intracellular signaling.

To prevent partial compensation and to test the feasibility of the AAVMYO-CRISPR/Cas9 system to delete several genes, we targeted both mouse *Acvr2a* and *Acvr2b* genes by simultaneously injecting two AAVMYO viruses (each targeting one gene with seven different sgRNAs (localization see Fig. S13a)) at a dose of $3 \times 10^{11}$ vg (each virus) into TA muscle of 8-week-old Cas9mKI mice (Fig. 7a) and analyzed mice 6 weeks later. Virus-injected muscles expressed only 18% of *Acvr2a* and 26% or *Acvr2b* transcripts compared to PBS-injected muscle (Fig. 7b), confirming successful targeting. We then used amplicon-NGS in combination with CRISPResso2 analysis[26], to determine DNA editing

efficiency for each sgRNA at its target locus and its CRISPOR-predicted[19] off-target site in TA muscle. The *Acvr2a* gene was significantly edited at the target site (Fig. S13b; Supplementary Data 1) and editing at *Acvr2b* gene reached between 10 and 30% (Fig. S13c; Supplementary Data 1). Targeting of the primary off-target sites was negligible with the exception of sgAcvr2b-4 where editing seemed even higher than at the target site (Fig. S13d, e; Supplementary Data 1).

Phenotypically, AAVMYO-7sgAcvr2a/b-injected mice gained significantly more body mass than controls (Fig. 7c) and muscles of the injected leg were 40–50% heavier than in PBS-injected mice (Fig. 7d), which is highly similar to the phenotype of *Acvr2a/b* double-knockout mice[32]. Like in experiments targeting *Musk*, the intramuscular AAV-MYO injection at this high dose caused a gain in muscle mass in the contralateral leg similar to that in the injected leg (Fig. S14a). Muscle growth in global *Mstn*-deficient mice is mediated via hyperplasia and hypertrophy[29], while myostatin signaling blockade after weaning (>3–4 weeks) predominately stimulates hypertrophy[28,33,34]. Consistent with these results, immunohistochemistry of TA muscles injected with AAVMYO-sgAcvr2a/b (Fig. 7e) and quantitative measurement of minimal fiber feret diameter[35] showed a consistent rightward shift in fiber size distribution and a significant increase in mean fiber size of all fiber types (Fig. 7f) without affecting fiber number (Fig. S14b). These results are highly consistent with those obtained with knockout mice and demonstrate the utility of AAVMYO-CRISPR/Cas9 to inactivate multiple genes and reproduce the phenotypes of traditional knockout mice, without the need to breed additional mouse lines.

## Discussion

This article presents a rapid and highly efficient tool to investigate the function of single or multiple genes in adult skeletal muscle fibers. Feasibility and efficiency of the system is demonstrated by knocking out essential genes for the integrity of the NMJ and for skeletal muscle fiber growth.

We show that high, long-term Cas9 expression in skeletal muscle fibers does not affect muscle size or function. Others have used AAV to deliver Cre to LSL-Cas9KI mice to excise the stop cassette and drive Cas9 expression[11,36]. While this approach allows for the use of Cas9-GFP as a transfection marker and reduces any potential side effects of prolonged Cas9 expression, AAV-mediated delivery of Cre would also lead to Cas9 expression in any AAV-targeted tissues, including the heart and liver. To our best knowledge, highly specific AAV-compatible promoters for skeletal muscle fibers do not currently exist. As such, our AAVMYO-CRISPR/Cas9mKI strategy represents a major advancement for somatic gene perturbation of mouse skeletal muscle fibers.

While CRISPR/Cas9 systems for somatic gene deletion have been described for some tissues, including brain and liver[11,37], such a versatile tool has so far been missing for skeletal muscle fibers. Previous work has demonstrated successful somatic gene editing using CRISPR in muscle stem cells (MuSCs) although with rather low efficiency[38,39]. While editing efficiency can be increased by sorting MuSCs based on a fluorescent transfection marker, this is not possible for multi-nucleated skeletal muscle fibers. Thus, the successful depletion of a gene by CRISPR in muscle fibers is only possible when indels are generated in both alleles in the majority of myonuclei. Such high efficiencies are not required in CRISPR/Cas9-mediated editing approaches that aim to correct gene mutations causing muscular dystrophies[38,40–44]. In these experiments, correcting the mutation in a subset of myonuclei and in one allele is sufficient as the corrected protein will distribute in a large part of the muscle fiber cytoplasm.

While CRISPR/Cas9-mediated gene deletion in mouse embryos has largely replaced the traditional stem cell-based gene targeting approaches as it shortened the time to create founder mice to a few weeks and created the possibility to target multiple genes simultaneously[45], it still requires many founder breedings with different mouse lines to eventually achieve the final genotype needed for a

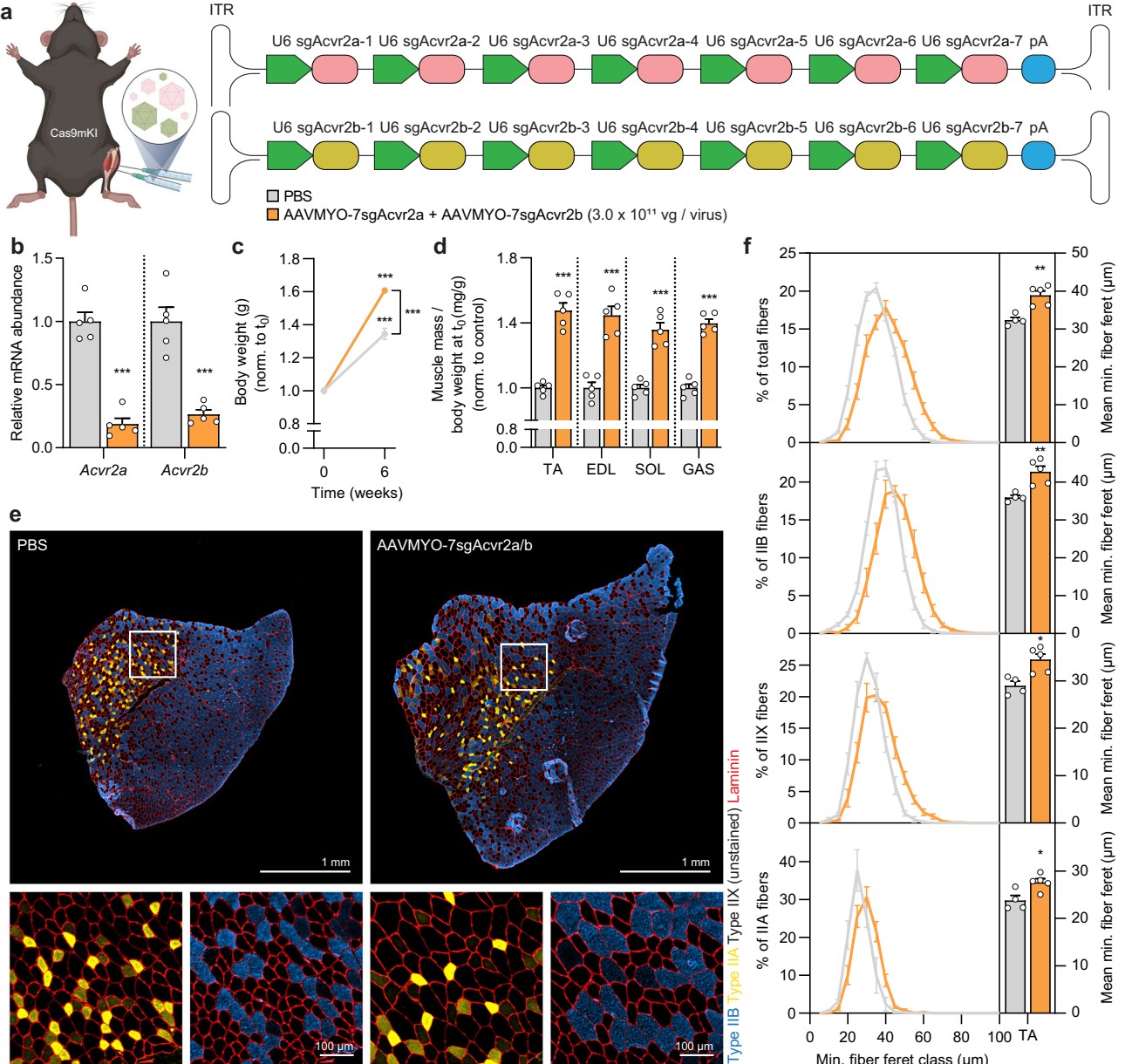

**Fig. 7 | AAVMYO-CRISPR/Cas9-mediated double knockout of Acvr2a/Acvr2b causes strong skeletal muscle fiber hypertrophy. a** Schematic representation of the experimental procedure. For abbreviations, see legend to Fig. 2. **b** Relative mRNA expression of Acvr2a and Acvr2b in *gastrocnemius* (GAS) muscle of Cas9mKI mice, injected with PBS or AAVMYO-7sgAcvr2a/b 6 weeks post-injection. **c** Body mass of Cas9mKI mice before and 6 weeks after intramuscular injection of PBS (gray) or AAVMYO-7sgAcvr2a/b (orange). **d** Mass of *tibialis anterior* (TA), *extensor digitorum longus* (EDL), *soleus* (SOL) and GAS muscles of the AAVMYO-7sgAcvr2a/b-injected (orange) or PBS-injected (gray) legs of Cas9mKI mice. **e** Representative images of TA cross sections stained with antibodies to type IIB (blue), type IIA (yellow), and laminin (red) from Cas9mKI mice injected as indicated. Note that type2X muscle fibers are not stained. **f** Total and fiber type-specific minimal fiber feret distribution (left) and mean minimal fiber feret (right) of TA muscle of Cas9mKI mice, injected with PBS (gray) or AAVMYO-7sgAcvr2a/b (orange). Data are means ± SEM. For (**b**–**d**), $n = 5$ mice. For (**e**) and (**f**), $n = 4$ (control) and 5 (AAVMYO-7sgAcvr2a/b) mice. Statistical significance is based on two-way ANOVA with Tukey's post-hoc test (**c**) and unpaired student's two-sided t-test (**b**, **d**, **f**). $*P < 0.05$, $**P < 0.01$, $***P < 0.001$. Experimental scheme was created with BioRender.com. Source data and precise *p*-values are provided as a source data file.

study. The method we established here allows to conditionally knock out single or multiple genes in muscle fibers without any prior breeding. The finding that the targeting of *Musk* and of the two myostatin/activin receptors *Acvr2a* and *Acvr2b* results in a phenocopy of knockout mice, provides strong evidence that the AAV-CRISPR/Cas9 method is as efficient as the Cre/loxP targeting system. However, it remains open whether this is also true for all genes. As is the case for the Cre/loxP system, the lack of a strong phenotype will require to assess the knockout efficiency with independent methods. We nevertheless hypothesize that the use of mice expressing Cas9 at high levels

in all muscle fibers in combination with AAVMYO to deliver sgRNAs strongly increase the likelihood to achieve efficient somatic gene deletion. Our method also allows for both, systemic or local (in a single muscle) gene editing, which may be essential in cases where a gene knockout causes severe morbidity or death. Thus, this method also contributes to the 3 R principle by strongly reducing the number of mice needed to investigate the function of genes in vivo.

The method was established by targeting PKCα, as work in the retina has provided evidence for an efficient knockout using CRISPR/Cas9[15] and based on the availability of high-quality antibodies to PKCα.

Using this target, we were able to optimize the delivery method for the sgRNA by using AAVMYO instead of AAV9. With this optimized set-up, the amount of PKCα was lowered by ~80% in the injected muscle. Interestingly, DNA editing of the *Prkca* locus, measured by TIDE analysis, did not reach 80% but was only 23%. Several reasons can account for the quantitative difference between genome editing and loss of protein. First and foremost, only ~50% of the nuclei in a muscle isolate are myonuclei[46,47]. The remaining nuclei are derived from mononucleated cells, such as FAPs, macrophages, MuSCs, endothelial cells, smooth muscle cells, or Schwann cells. All these non-muscle fiber cells do not express Cas9 in the Cas9mKi mice and are not edited. Hence, even 100% DNA editing in muscle fibers will be detected as ~50% in a muscle lysate. CRISPR/Cas9 editing in cultured C2C12 myotubes using the same sgRNA resulted in 60% editing and a reduction of the protein by more than 90%. With this in mind, the ~50% editing in myonuclei correlates well with the 80% reduction in protein. Another possible contributor to the lower editing efficiency in skeletal muscle fibers than in C2C12 myotubes could be a different chromatin environment of the target site in the *Prkca* locus[48].

We also detected ~5% *Prkca* gene editing in the heart upon systemic delivery of AAVMYO-3sgPKCα-1. The most probable reason for this is the scattered expression of Cre in the developing heart[12], which may result in some excision of the stop cassette in the Rosa26[Cas9-EGFP] knockin mice. Consistent with this notion, very low levels of Cas9 were detected in the heart by Western blot analysis (see Fig. 1c). We do not know whether the low editing efficiency of 5% is sufficient to create *Prkca* knockout cardiomyocytes but this seems rather unlikely. Moreover, *Prkca*, like the other genes targeted in this study, is not essential for cardiomyocyte function. Importantly, *Prkca* gene editing could not be detected in the liver as this tissue does not at all express the Cas9 protein. Thus, the here presented AAVMYO-CRISPR/Cas9 gene perturbation tool is very specific to interrogate gene function in skeletal muscle fibers with the caveat that some indels may also be created in the heart. As HSA-driven Cre expression was reported for the developing heart, the tamoxifen-inducible iCas9mKI mice might be a way to overcome this potential challenge.

As a functional proof-of-concept, we perturbed MuSK function, which is essential for NMJ formation and maintenance[21]. *Musk* expression in adult mice is confined to sub-synaptic nuclei, which lay directly underneath the NMJ. Sub-synaptic *Musk* expression is based on local, NMJ-derived signals that overwrite activity-mediated transcription suppression in non-synaptic myonuclei[21]. Denervation and hence loss of electrical activity results in *Musk* re-expression in non-synaptic myonuclei. Thus, efficient editing in myonuclear DNA is important to abrogate *Musk* expression in in muscle fibers. At the highest AAVMYO dose, *Musk* transcripts were reduced up to 98%. The reason for this strong loss of *Musk* expression is based on the use of multiple sgRNAs that introduce large deletions between different target sites, which are likely to de-stabilize mRNA. Indeed, PCR amplification of cDNA derived from polyA-positive mRNA showed the presence of many truncated products between the individual sgRNA-target sites (Fig. S11). We also detected many short indels for each sgRNA in *Musk* (Fig. S10), which may result in frameshifts and the occurrence of premature termination codons that cause nonsense-mediated mRNA decay. A strong reduction of transcript levels was also observed for *Acvr2a* and *Acvr2b* using multiple sgRNAs. We also tested whether the use of multiple sgRNAs would increase off-target editing. Out of the 22 primary off-target sites tested, only two exhibited rather high DNA editing. However, both loci are in intronic or intergenic regions and hence should not impair function of nearby genes. Moreover, the somatic gene perturbation studies shown here did not last more than a few weeks, which makes it unlikely that off-target editing might be responsible for any of the observed effects. We hence advise to not use sgRNAs for which off-target sites are located in exons and to confirm the results of perturbation experiments for genes with

unknown function with a second, independent set of sgRNAs targeting the same gene.

Although the focus of our work was to use the MuSK knockdown as a proof-of-principle to demonstrate efficiency of the method, our data also show that MuSK is essential for the maintenance of the NMJ in the adult and that it is critical for muscle mass maintenance. This has so far only been shown indirectly by (i) injection of the MuSK ectodomain into adult mice that triggered the production of autoimmune antibodies and resulted in the deterioration of the NMJ reminiscent of myasthenia gravis[49], (ii) local shRNA-mediated suppression of *Musk* by electroporation, which led to NMJ loss[4] and (iii) by conditionally deleting *Musk* in muscle fibers by muscle creatine kinase-driven Cre, which caused death of the mice at ~1 month of age[23].

AAVMYO-CRISPR/Cas9 knockdown of *Musk*, *Acvr2a,* and *Acvr2b* when injected systemically or at the highest dose into TA muscle resulted in a systemic loss of the targeted proteins. The systemic effect of the high intramuscular doses is likely based on the body-wide spreading of the sgRNA-expressing recombinant viruses via the blood stream and the subsequent transduction of skeletal muscles. Systemically administered AAVMYO targets all muscles but has the highest transduction rate in the diaphragm (Fig. 4c, refs. 7,8). While such body-wide spreading may not be a problem for most experiments, in case of *Musk* deletion, NMJs deteriorate and muscles become denervated[23]. At the highest dose of $3 \times 10^{11}$ vg/mouse (corresponding to ~$1.3 \times 10^{13}$ vg/kg), mice started to lose weight 14 days post-injection and reached euthanization criteria after 3 weeks (Fig. 6b). Examination of the diaphragm muscle indicated NMJ deterioration. Based on this, mice injected with the highest dose needed to be analyzed already at 3 weeks post-injection, which explains the less severe phenotype in the hindlimbs. Lowering the dose of the injected virus to 3 or $1 \times 10^{10}$ vg/mouse largely prevented weight and muscle mass loss in the contralateral leg while the injected muscle still showed all signs of NMJ deterioration and denervation. As finding the correct amount of AAV that perturbs the function locally but does not spread systemically might be challenging, we also used AAV9 in combination with neuraminidase to deliver the seven sgRNAs targeting *Musk*. Indeed, even at a high dose of $3 \times 10^{11}$ vg/mouse, loss of the NMJ and subsequent loss of muscle mass remained restricted to the injected hindlimb muscles (Fig. S12). In summary, the here presented AAV9/AAVMYO-CRISPR/Cas9 systems are very versatile for the local and systemic perturbation of gene function in skeletal muscle fibers.

We also demonstrate efficacious, simultaneous inactivation of multiple genes (*Acvr2a* and *Acvr2b*) with the AAVMYO-CRISPR/Cas9 system, opening the possibility of studying several genes or signaling pathways concurrently. Although our experiments targeting *Prkca* indicate that one sgRNA can be sufficient to eliminate a gene, testing each sgRNA in vitro prior to in vivo application is laborious. We find that the targeting efficiency of individual sgRNAs varies in myonuclei between approximately 20% to 60% (Figs. S10 and S13) and that the use of multiple sgRNAs generates bigger deletions (Fig. S11). Hence, we suggest targeting each gene with several different sgRNAs, minimizing the risk of insufficient protein loss. Since one AAV has sufficient packaging capacity for at least 7 sgRNAs, we anticipate that up to three genes could be silenced with one AAV. By delivering two AAVs (as done here for *Acvr2a* and *Acvr2b*), up to six independent genes could be silenced simultaneously, allowing interrogation of entire signaling pathways, specifically in skeletal muscle fibers.

In summary, we conclusively demonstrate that AAV9- or AAVMYO-mediated delivery of sgRNA to Cas9-expressing skeletal muscle fibers allows fast, efficient, and specific gene knockouts. The multiplexable nature and capacity to induce systemic or local gene editing further strengthens the universality of the system. Therefore, this system provides an invaluable resource to perform loss-of-function studies in skeletal muscle fibers compared to traditional knockout mouse models and promises to greatly accelerate the

interrogation of novel gene targets with a much reduced number of animals needed and thus will strongly contribute to our understanding of skeletal muscle biology.

# Methods

## Ethical statement

All procedures involving animals were performed in accordance with Swiss regulations and approved by the veterinary commission of the canton Basel Stadt.

## Mice

Mice were kept on a 12 h light-dark cycle (6 am to 6 pm) at 22 °C (range 20–24 °C) and 55% (range 45–65%) relative humidity. Cas9 knockin mice[11] were crossed with HSA-Cre[12] or HSA-Mer-Cre-Mer mice[13] to generate Cas9mKI or iCas9mKI, respectively. Littermates, knockin for Cas9 but not expressing Cre recombinase, were used as controls. Cas9mKI/iCas9mKI mice and WT controls were bred on a C57BL/6JRj background. For AAV administration, only mice older than 6 weeks were selected, while sex was not considered in this study.

## Cell culture C2C12

Murine C2C12 myoblasts (CRL-1772, ATCC) were cultured in growth medium (DMEM (Gibco) supplemented with 10% fetal bovine serum (Biological Industries) and 1% penicillin/streptomycin (Sigma)) at 37 °C in an atmosphere of 5% $CO_2$. After reaching 70% confluence, cells were transiently transfected using Lipofectamine 2000 (Invitrogen), according to the manufacturer's protocol. At 48 h post-transfection, cells were incubated in growth medium, supplemented with 3 μg/ml puromycin (Sigma), for another 48 h to select for transfected cells. After selected cells reached confluence, cells were incubated in differentiation medium (DMEM (Gibco) supplemented with 2% horse serum (Biological Industries) and 1% penicillin/streptomycin) for 5 days to induce formation of multi-nucleated myotubes.

## AAV administration

Prior to AAV administration, mice were anesthetized by isoflurane inhalation. For intramuscular injection, the TA or TA and GAS muscle of adult mice (older than 6 weeks) was injected with 50 μL of AAV ($3 \times 10^{11}$ vg, if not stated differently) in PBS. For intravenous injection, 100 μL of AAV ($1 \times 10^{14}$ vg/kg) in PBS were injected into the lateral tail vein of 6-week-old mice. For targeting of PKCα or Acvr2a/Acvr2b, PBS or non-targeting AAV-injected control or Cas9mKI mice were used as control. For targeting of *Musk*, PBS-injected Cas9mKI mice or AAVMYO-7sgMusk-injected control mice were used as control.

## Cardiotoxin injury

14-week-old mice were anesthetized by isoflurane followed by an injection of 50 μL cardiotoxin (10 μM in 0.9% NaCl) into the left TA muscle. The right TA muscle was sham-injected with 0.9% NaCl and was used as uninjured control. Mice were analyzed 21 days post-injection.

## Denervation

Mice were anesthetized by isoflurane inhalation 6 weeks post-AAV administration. After making a small incision on the skin between sciatic notch and knee, the sciatic nerve was exposed by gentle separation of muscles under the skin. The nerve was then lifted using a glass hook and disrupted by removing a 5 mm piece. The wound was closed by surgical clips and mice were returned to their cage. Mice were treated with Buprenorphine (0.1 mg/kg of body weight) 1 h before and for 2 days after operation.

## sgRNA design and AAV vectors

The sgRNAs, listed in Supplementary Data 2 were selected using CRISPOR v5.01[19] to minimize off-target effects and assembled as previously described using the multiplex CRISPR/Cas9 assembly kit[50]. An array of three to seven human U6/sgRNA cassettes were cloned into an AAV transfer vector. The AAV transfer vectors used for 3-plex sgRNA delivery into skeletal muscle were cloned between AAV serotype 2 ITR's including a cloning site for multiplexed hU6-sgRNA insertions (MluI and KpnI (NEB)), the ubiquitous CMV promoter, tdTomato, WPRE and bovine growth hormone polyA signal. For 7-plex sgRNA delivery by AAV, the CMV-tdTomato-WPRE sequence was removed from the AAV transfer vector.

For in vitro CRISPR applications, sgRNAs were cloned into an all-in-one CRISPR/Cas9 vector using BbsI (NEB). The all-in-one CRISPR/Cas9 vector was cloned between AAV serotype 2 ITRs including a human U6 promoter, sgRNA scaffold, an EFS promoter, SpCas9 linked to puromycin N-acetyltransferase via a GSG-P2A linker and bovine growth hormone polyA signal. Complete vector maps and sequences are available upon request.

## AAV production, purification, and titration

The AAV-sgRNA plasmid vectors were used for AAV production and purification. Briefly, adherent HEK293T cells (CRL-3216, ATCC) were transiently transfected with transfer (AAV-sgRNA construct), AAV helper (AAV9 (a gift of J. M. Wilson Addgene, plasmid # 112867), AAVMYO[8], AAVMYO2[7], or AAVMYO3[7]) and pAdDeltaF6 helper (a gift from J. M. Wilson Addgene, plasmid # 112867) plasmid using PEI MAX (Polyscience). For small or large AAV preparations, ten or twenty HEK293T confluent 15-cm tissue culture plates were processed, respectively. The supernatant was collected 48 and 72 h post-transfection and cells were dislodged 72 h post-transfection in PBS. Cells were centrifuged at $500 \times g$ at 4 °C for 10 min and resuspended in AAV lysis solution (50 mM Tris-HCl, 1 M NaCl, 10 mM $MgCl_2$, pH 8.5). 50 U of salt active nuclease (Sigma) was added per harvested 15 cm dish and incubated at 37 °C for 1 h with continuous shaking. The lysate was spun at $4000 \times g$ at 4 °C for 15 min and supernatant was collected. AAV particles from the supernatant were precipitated by adding polyethylene glycol 8000 (Sigma) to a final concentration of 8% (w/v), incubated for 2 h at 4 °C and then spun at $4000 \times g$ at 4 °C for 30 min. The supernatant was discarded, while the pellet was resuspended in AAV lysis buffer and pooled with the cell lysate. AAV particles were purified by using a 15–25–40–60% iodixanol (Serumwerk) gradient. The gradient was centrifuged at $292,000 \times g$ (Beckman type 70 Ti rotor) for 2 h at 4 °C and the AAV particles were collected from the 40–60% phase interface. The extract was passed through a 100 kDa MWCO filter (Millipore) and washed with PBS supplemented with 0.01% Pluronic F-68 surfactant (Gibco) until buffer was exchanged completely. The final volume was decreased to reach a final AAV concentration of >$1 \times 10^{13}$ vg/ml. Virus was titered using RT-qPCR targeted to the ITRs, as previously described[51], using a PvuII (NEB)-linearized plasmid standard. Primers used for titration are listed in Supplementary Data 2.

## Protein isolation and Western blot analysis

Dissected muscles were snap-frozen in liquid nitrogen and pulverized. Proteins were extracted using RIPA lysis buffer (50 mM Tris-HCl pH 8.0, 150 mM NaCl, 1% NP-40, 0.5% sodium deoxycholate, 0.1% SDS) supplemented with protease and phosphatase inhibitors (both Roche) for 2 h at 4 °C, followed by sonication. Lysates were centrifuged at $16,000 \times g$ for 20 min at 4 °C and Pierce BCA Protein Assay Kit (Thermo Fisher Scientific) was used to determine protein concentration. Equalized protein samples were separated on 4–12% Bis-Tris Protein Gels (NuPage Novex), followed by transfer to nitrocellulose membranes (GE Healthcare Life Science). Membranes were blocked for 1 h by 5% BSA in PBS-T (1% Tween-20) and incubated with primary antibody in blocking solution overnight at 4 °C. After 3 washes with PBS-T, membranes were incubated with secondary horseradish peroxidase-conjugated antibody for 1 h at RT. After washing 3 times with PBS-T,

proteins were visualized using KPL LumiGLO (Seracare) and chemiluminescence was captured by a Fusion Fx machine (ViberLourmat). Protein abundance was quantified using the FusionCapt Advance software with linear background subtraction. Used antibodies are listed in Supplementary Data 2.

## Immunostaining of muscle cross sections and muscle histology analysis

Animal tissue was dissected, prepared, and sectioned for immunohistochemistry as previously described[52]. Tissue for analysis of cytosolic expression of GFP or tdTomato was directly fixed in ice-cold 4% PFA (Electron Microscopy Science) for 2 h at 4°, followed by dehydration with 20% sucrose (Sigma) in PBS at 4 °C overnight. The next day, tissue was processed like non-fixed tissue as previously described[52]. TA muscle cross sections were blocked and permeabilized for 30 min at RT with 3% BSA, 0.5% Triton X-100 in PBS. Primary antibodies were diluted in blocking solution for 2 h at RT. Sections were washed with PBS three times before being incubated in secondary antibody solution for 1 h at RT. All antibodies are listed in Supplementary Data 2. Sections were washed with PBS four times and mounted with ProLong Gold Antifade Mountant (Invitrogen). Muscle sections were imaged at the Biozentrum Imaging Core Facility with a SpinD confocal microscope (Olympus). The previously described script for automated muscle cross section analysis[53] was further developed in-house and is available on GitLab (https://git.scicore.unibas.ch/imcf/myosoft-imcf).

For muscle histology analysis, muscle cross sections were fixed for 10 min in 4% PFA followed by hematoxylin (Merck) and eosin staining (Merck). Samples were dehydrated by serial washes in 100% EtOH and Xylene and mounted using DPX (Sigma). An Olympus iX83 microscope with color camera was used to acquire images.

## MuSK and AChR density quantification

GAS muscle was prepared for immunohistochemistry and stained as described above. Z-stack images were acquired on a SpinD confocal microscope (Olympus) using a ×60 objective. SV2 staining was used to identify NMJ regions. Images were analyzed on ImageJ software following the workflow for AChR density quantification, as previously described[52]. SV2 staining was used to define the region of interest (NMJ). The same workflow was adapted to quantify MuSK density. Integrated density values were normalized to NMJ size and further normalized to control for visualization.

## Whole-mount NMJ staining

EDL muscles were fixed, cut into bundles and prepared for NMJ staining as previously described[52]. The presynapse was visualized using a primary antibody mix against neurofilament and synaptic vesicle protein, while the postsynapse was stained using A647-conjugated α-bungarotoxin. MuSK was visualized using specific primary antibodies[54]. NMJs were imaged at the Biozentrum Imaging Core Facility with a SpinD confocal microscope (Olympus).

## In vitro muscle force

Fast-twitch EDL and slow-twitch SOL muscles were carefully isolated for in vitro force and fatigue test as previously described[52]. The measurement was carried out on the 1200 A Isolated Muscle System (Aurora Scientific) in Ringer solution (137 mM NaCl, 24 mM NaHCO₃, 11 mM glucose, 5 mM KCl, 2 mM CaCl₂, 1 mM MgSO₄, 1 mM NaH₂PO₄) which was gassed with 95% $O_2$, 5% $CO_2$ and kept at 30 °C.

## Genomic DNA isolation, PCR amplification, and TIDE

Cells were washed in PBS, while dissected tissue was snap-frozen in liquid nitrogen and pulverized in liquid nitrogen. Genomic DNA from cells or tissue was isolated using the DNeasy blood and Tissue kit (Qiagen) according to the manufacturer's protocol. DNA was amplified

using standard PCR using LongAmp Taq DNA polymerase (NEB) targeting the sgPKCα-1 editing site with a 200–500 bp-long amplicon. Used primers are listed in Supplementary Data 2. PCR amplicons were purified using AMPure XP beads (Beckman) and Sanger-sequenced (Microsynth) using one of the two PCR primers. TIDE was applied to sequencing chromatograms to assess CRISPR-editing efficiency of the target locus[55].

## Amplicon deep sequencing analysis

PCR of genomic DNA was performed using Q5 DNA polymerase (NEB) and primers (Supplementary Data 2) designed against an amplicon of 150–600 bp targeting the CRISPR-locus or the CRISPOR-predicted off-targets[19] (Table S1). First-round PCR primers contained adapter sequence for DNA/RNA UD Indexes (Illumina). The second round of PCR and pooling of samples was performed according to the Illumina Nextera DNA library preparation guide. Pooled libraries were sequenced with 20% PhiX spike-in with standard 500/600 cycles kit PE 2x251/2x301 on an Illumina MiSeq instrument. Samples were demultiplexed according to assigned barcodes and FASTQ files were analyzed using the CRISPResso2 software package 2.2.12[26].

## CRISPR/Cas9-mediated large deletion analysis

*Musk* cDNA was PCR amplified using Q5 DNA polymerase (NEB) and primers flanking sgRNA(1–7) (Supplementary Data 2). PCR products were cleaned up using AMPure beads (Beckman Coulter) in a 1:2 (input:beads) ratio. They were analyzed on a fragment analyzer (Agilent), fragmented and barcoded using NEBNext® Ultra™ II FS DNA Library Prep Kit (NEB). Libraries were pooled and sequenced with 10% PhiX spike in with a 500 cycle MiSeq Reagent Nano Kit v2 (Illumina). Samples were demultiplexed according to assigned barcodes and FASTQ files were aligned to the genome using STAR 2.7.9[56]. Sashimi plots were created using the Integrative Genomics Viewer 2.16.1 (IGV) desktop application[57]. Read junctions with a lower abundance than 0.5% of total reads were excluded from illustration in Fig. S11.

## AAV genome copy number quantification

The AAV viral genome copy number per nuclei was determined by PCR using PowerUp SYBR Green Master Mix (Applied Biosystems) and primers (Supplementary Data 2) targeting the tdTomato-WPRE or ITR sequence (AAV) and the R26 locus (nuclei) on a QuantStudio5 (Applied Biosystems) instrument. The cycle threshold (CT) values were converted into copy numbers by measuring against a standard curve of the AAV transfer plasmid or the Ai9 plasmid (a gift from H. Zeng Addgene plasmid # 22799). The AAV genome copy number was divided by the number of R26 copies (nuclei) to normalize for tissue input.

## RT-qPCR

Pulverized muscle tissue was lysed in RLT buffer (Qiagen) and RNA was extracted using the RNeasy® Mini Kit for fibrous tissue (Qiagen). cDNA was reverse transcribed using the iScript™ cDNA synthesis kit (Bio-Rad) and 500 ng of RNA according to the manual. RT-qPCR was performed using PowerUp SYBR Green Master Mix (Applied Biosystems) and target-specific primers (Supplementary Data 2) on a QuantStudio5 (Applied Biosystems) instrument. Data were analyzed using the comparative CT method ($2^{-\Delta\Delta Cq}$). Raw CT values of targets were normalized to CT values of a housekeeper (Tata-box-binding protein), which was stable between conditions, and then further normalized to the control group for visualization.

## Statistics and reproducibility

All values are expressed as mean +/− SEM, unless stated otherwise. Data were analyzed in GraphPad Prism 8.0.2. Unpaired student's two-sided t-test were used for pairwise comparison. One-way ANOVAs with Fisher's LSD post-hoc tests were used to compare between three groups, while Tukey post-hoc tests were used for comparison

between more than three groups, as long as the ANOVA reached statistical significance. A two-way ANOVA with Tukey's post-hoc test was used to compare between multiple groups over different conditions. Significant differences ($*P < 0.05$, $**P < 0.01$, $***P < 0.001$) are reported on figures, where appropriate. For images and blots, a representative sample was selected from all indicated samples for representation.

### Reporting summary
Further information on research design is available in the Nature Portfolio Reporting Summary linked to this article.

## Data availability
Source data, full uncropped Western blots and precise $p$-value are provided as Source data file with this paper. Deep sequencing data generated in this study have been deposited in the National Center for Biotechnology Information (NCBI) Sequence Read Archive (SRA) under accession code PRJNA1017672. Data are available from the authors on request.

## Code availability
The previously described script for automated muscle cross section analysis (Myosoft)[53] was further developed in-house and is available on GitLab (https://git.scicore.unibas.ch/imcf/myosoft-imcf).

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

## Acknowledgements

We thank the Biozentrum core facilities for their technical support with imaging (Imaging Core Facility), sequencing (Genomics Facility), computing (SciCORE) and mice housing (Animal Facility). We thank Dr. D. J. Ham for his comments on the manuscript. This work was supported by funds from the Swiss National Science Foundation (#189248) and the cantons of Basel-Stadt and Basel-Landschaft awarded to M.A.R.

## Author contributions

Conceptualization: M.T. and M.A.R.; methodology: M.T.; investigations: M.T., S.L., and F.O.; statistical analysis: M.T.; visualization: M.T.; supervision: M.A.R., R.J.P., and D.G.; writing original draft: M.T. and M.A.R.; writing review and editing: M.T., D.G., and M.A.R.; funding acquisition: M.A.R.

## Competing interests

The authors declare no competing interests.
