## [Peer Review File · Nature Communications]

Reviewers' Comments:

Reviewer #1:

Remarks to the Author:

This manuscript describes a method to knock down genes in adult mouse skeletal muscle using AAV-CRISPR/Cas9 where Cas9 is either constitutively (Fig 1, S1) or inducibly (Fig S2) expressed in skeletal muscle and specific genes are targeted using sgRNAs delivered via AAV. They show knock down of PKCalpha (70-80% reduction in protein, Figs 2-4, S3-7), MusK (~90% reduction in mRNA, Figs 5-6, S8), and two myostatin receptors Acvr2A and Acvr2B (~80% reduction in mRNA, Figs 7, S9) and observe phenotypes in the latter 2 knock down animals consistent with those observed in conditional knockout models. They conclude that they have established a method to rapidly and efficiently elucidate the functional relevance of individual genes or entire signaling pathways in adult skeletal muscle without the prerequisite of generating more classical knockout mice.

I'm less excited than the authors about the broad applicability of the method because they targeted 4 genes and all 4 were knocked down, but none were knocked out. Thus, using this method to interrogate the function of novel genes might reveal that a gene is important for some aspect of muscle function as long as ~80% reduction in transcript is sufficient, or it might yield a false negative outcome for an essential muscle gene where 20% of WT expression is sufficient.

I'm also not convinced that constitutive (or even inducible) expression of Cas9 is without physiological consequence in adult skeletal muscle. The authors only looked at the weights of TA, EDL, GAS, SOL and Quad, fiber size and type only in TA, and contractile function only in isolated EDL without any experiments that stress the animals by exercise or aging so it may not be clear if phenotypes revealed by knocking down a gene are compounded by any undetected stress imposed by high Cas9 expression.

Other Issues

Figs 1C and S2: The GAPDH immunoreactivity is so highly variable as to render it unsuitable as a loading control.

Fig 2: The data do not support the legend title claiming knock out of Prkca. The authors should also reconsider whether the dots shown for quantitation of PKCalpha immunoreactivity in panel G align with the western blot shown in panel F.

Lines 313-315: The authors should read A Morin et al., 2023 PNAS.

Reviewer #2:

Remarks to the Author:

The authors have designed a clever method for rapidly testing gene knockdowns/knockouts in skeletal muscle using constitutively or inducibly activated Cas9 in conjunction with AAV(s) to deliver guide RNAs. They demonstrate their potential scientific value by knocking down four targets: Prkca as proof of concept; MuSK to demonstrate the ability to generate a muscle phenotype after a single gene knockdown; and Acvr2a and Acvr2b to demonstrate the ability to generate a muscle phenotype after multi-gene knockdown. While systemic knockdowns were highly successful, individual limb knockdowns were somewhat stymied by high knockdown efficiencies in the rest of the animal due to AAV leakage.

It is likely that these animals will prove highly useful to the field. While a direct comparison to shRNA based knockdowns isn't possible due to variability of efficiencies based on the particular sequences of shRNAs and gRNAs, the data shown here suggest that is a viable replacement and probable improvement on such established methods.

I recommend the paper be accepted after the authors make the following minor revisions:

Line 70-72: The wording of the sentence leaves unclear the fact that the Cre is expressed from a

muscle specific promoter (HSA). I would rewrite the sentence to include that information. As it stands it could be misinterpreted that only the tamoxifen induced Cre is muscle-specific.

Line 173: The figure being references is S7A, not S6A.

Line 343: It's not clear to me how the authors came to the 75% number. If 60% editing in vitro led to >90% knockdown, and the knockdown here is 80% (suggesting less editing), why do the authors argue that real editing must be 75%?

Line 385: The authors should more explicitly acknowledge that they struggled to balance high levels of local knockdown with systemic leakage. While they could lower the dose of AAVMyo to a point with minimal leakage, local knockdown efficiencies were likewise reduced. For certain genes, this may not be limiting, but for others it could be. While AAVMyo has obvious advantages over AAV9 for systemic delivery, AAV9 may actually be better for individual limb or muscle knockdown experiments based on the authors' data, which showed AAV9 had significantly less transduction due to leak. Additionally, it's possible that local AAV administration could possibly be coupled with local tamoxifen administration in the iCas9mKI mice. Future studies should further refine this method.

Figure S3C: I am curious as to the reason for the reduced Cas9 expression in the C2C12s that received a construct containing the targeting guide RNAs (vs sgNT). If the authors have a theory as to why, it would be nice to have an explanation in the discussion.

Signed,
Julie Crudele

Reviewer #3:

Remarks to the Author:

In Thürkauf et al.'s manuscript, "Fast, multiplexable and highly efficient somatic gene deletions in adult mouse skeletal muscle fibers using AAV-CRISPR/Cas9 " the authors describe CRISPR/Cas-based gene editing in skeletal muscle fibers in muscle-specific Cas9-expressing mice using muscle-targeting AAVMYO-sgRNA.

In particular, the authors demonstrate a method to target genes specifically in adult skeletal muscle fibers after intramuscular or systemic delivery of AAV-sgRNA.

The authors also demonstrated that use of AAVMYO in mouse muscles resulted in higher transduction efficacy compared to conventional AAV-sgRNA. Overall, the results are interesting. However, there are several critical issues that should be addressed.

Major comments

The authors highlight that muscle-specific editing can be achieved in Cas9 and Cre-expressing mice. However, this study lacks supporting data to substantiate the specificity of the system. The authors should prove evidence that systemic or intramuscular delivery of AAV-sgRNA did not trigger genome editing in the heart or liver, as these organs do not express Cas9 protein in this particular mice.

(In Figure 2, the liver exhibited higher levels of AAV genome compared to TA or heart following intramuscular delivery of AAV. However, the DNA editing was only analyzed in TA muscles, and a comparison of gene editing among these three muscles was not conducted. In Figure 4, as well as in other data, similar results were observed exclusively in skeletal muscles without any comparative analysis with the heart or liver.

1. In this study, the authors propose that crossing the Cre-dependent Rosa26 with mice that express Cre resulted in muscle-specific Cas9 expression, while the heart and liver did not show such expression. The authors clearly demonstrate how muscle-specific expression is induced in these mice.
2. In Figure 2A, the authors should provide evidence to support that using three copies of sgPKCa-1 leads to improved genome editing efficiency compared to a single sgRNA treatment group.
3. In line 106, the authors mentioned that co-injection of neuraminidase with AAV resulted in

improved AAV transduction in skeletal muscle. However, the authors should provide evidence by comparing AAV transduction with and without neuraminidase to support this conclusion.

4. In Figure 6 and 7, it is unclear what the indel frequency is for each of the sgMUSK-1 to 7 and sgACvr2a/b. Further clarification on the genome editing efficiency for each specific sgRNA is needed.

In addition, did the use of the sgRNA cocktail result in large gene deletions instead of small indels? The authors should provide data on the mutation patterns to demonstrate the effects of the sgRNA cocktail on genome modifications.

5. In Figure 5, it is not clear what the relative Musk expression and genome editing frequency in other muscles, such as GAS, EDL, SOL, TRI and MAS.

6. In Figure S8, what is the reason behind the increased Musk expression in the group treated with AAVMYO-7sgMusk (1x10¹¹vg)?

7. In Figure 6E, it is observed that the level of postsynaptic AChRs appears to be decreased in the low dose group of AAV-treated mice compared to the high-dose group. The authors should provide quantitative data on the AchR level.

Further clarification is needed to understand the lack of correlation between low expression of Chma1 etc and the dose of AAVMYO-7sgMusk.

8. The authors hypothesize that local injection of AAVMYO in the TA muscle may affect other muscles due to systemic circulation of AAV. To confirm this, the authors should quantify the viral genome in the blood after intramuscular injection of AAV, in order to provide evidence for their speculation.

9. The authors should provide the mismatched sequence information at the potential off-target sites. Additionally, it is important to analyze the off-target effects following systemic delivery, particularly in the highest dose of AAVMYO, to comprehensively assess the potential off-target effects of the treatment.

Minor

1. The authors should perform a statistical comparison of the data between AAV9 and AAVMYO (Fig 3D-G, Fig 4F-G, Fig S5A), as well as between different disease dose (Fig 6D; 1x10¹¹vg vs 3x10¹¹vg), to thoroughly assess any differences and provide a comprehensive analysis.

2. Please provide a statistical analysis to specify the difference observed in Figure S9B.

Dear Cara,

We submit the revised version of our paper entitled “Fast, multiplexable and highly efficient somatic gene deletions in adult mouse skeletal muscle fibers using AAV-CRISPR/Cas9”. As delineated below, we have addressed all of the comments raised by the three reviewers. We would like to thank the reviewers as we think that the new experiments have further strengthened the paper.

REVIEWER COMMENTS

Reviewer #1 (Remarks to the Author):

I’m less excited than the authors about the broad applicability of the method because they targeted 4 genes and all 4 were knocked down, but none were knocked out. Thus, using this method to interrogate the function of novel genes might reveal that a gene is important for some aspect of muscle function as long as ~80% reduction in transcript is sufficient, or it might yield a false negative outcome for an essential muscle gene where 20% of WT expression is sufficient.

We thank the reviewer for this comment but respectfully disagree. The fact that the phenotypes for *MuSK* and *ACVR2RA/B* are the same as those observed in the conditional knockout mice, is strong evidence that we lose the function of the targeted genes. We have now also analyzed the transcripts that remain expressed in the AAVMYO-7sgMusk CRISPR/Cas9 experiment and we detect *Musk* transcripts that contain large deletions (Fig. S11). We therefore assume that the strong drop in the amount of *Musk* mRNA is based on nonsense-mediated mRNA decay and similar mechanisms. We now also analyzed the mutations induced by the different sgRNAs and find DNA editing between 10 and 30% at each sgRNA target site (Fig. S10, Fig. S13). Thus, each allele in the myonuclei is likely edited at multiple sites by the 7 different sgRNAs.

We would also like to point out that the same argument can be made when creating conditional knockout mice with the Cre/loxP system. To the best of our knowledge, it is rare that there is 100% recombination seen. This is particularly difficult to show in skeletal muscle as it consists of a multitude of different cell types.

I’m also not convinced that constitutive (or even inducible) expression of Cas9 is without physiological consequence in adult skeletal muscle. The authors only looked at the weights of TA, EDL, GAS, SOL and Quad, fiber size and type only in TA, and contractile function only in isolated EDL without any experiments that stress the animals by exercise or aging so it may not be clear if phenotypes revealed by knocking down a gene are compounded by any undetected stress imposed by high Cas9 expression.

We added now several data to strengthen the point that overexpression of Cas9 in skeletal muscle fibers does

not affect muscle physiology and function. The added data on muscle force in the slow-twitch *soleus* muscle (new Fig. S1E-G), *in vivo* muscle function by measuring grip strength (new Fig. 1G), and challenging muscle by inducing muscle injury by injection of cardiotoxin followed by regeneration (new Fig. S1I). In all of those measurements, we could not detect any difference to the controls. We also would like to emphasize that marker mice (e.g., expressing GFP or tdTomato) use the exact same system (GFP or tdTomato knocked into the *Rosa26* locus and expression driven by lox-stop-lox-CAG promoter). These mice are widely used in muscle research and one could also question that such mice are physiologically the same as controls.

Figs 1C and S2: The GAPDH immunoreactivity is so highly variable as to render it unsuitable as a loading control.

We agree with this comment as we also realize this big difference in GAPDH levels between muscle, heart and liver. We therefore always loaded the same total protein amount. We now also tested additional loading controls: α -tubulin, histone H3 and GAPDH (see Reviewers’ Figure) but none of these proteins was really stable in all the tissues. We thus decided to only show the Ponceau staining of the blot to “normalize” protein loading.

Reviewers’ Figure: Comparison of housekeeping proteins to normalize protein loading from lysates from different tissues. Although the same amount of total protein was loaded, as visualized by Ponceau staining, the amount of α -tubulin, GAPDH and histone H3 varies greatly between tissues.

Fig 2: The data do not support the legend title claiming knock out of Prkca. The authors should also reconsider whether the dots shown for quantitation of PKC α immunoreactivity in panel G align with the western blot shown in panel F.

To avoid any overstatement, we changed the title of the figure to “Loss of PKC α in TA muscle...” We have carefully re-measured the protein abundance on Western blot shown and compared with the quantification. The quantification turns out to be correct.

Lines 313-315: The authors should read A Morin et al., 2023 PNAS.

We thank the reviewer for the hint. In fact, we are well aware of this important work providing strong evidence that expression of *Dmd* at high levels in a few myonuclei does not result in a homogenous distribution of dystrophin and hence rather low expression of *Dmd* in all myonuclei would be preferred. In our discussion we only refer to the fact that CRISPR/Cas9 editing of a mutated gene does not require the same efficiency as the knocking down/out of a gene of interest. For example, if mutated *Dmd* is edited in all myonuclei in one allele only, the expression of dystrophin would be around 50%, which is certainly sufficient to restore its function. However, the knocking out a gene of interest in only one allele would rarely result in a phenotype as heterozygous knockout mice usually do not have a phenotype. As our work shows that we can phenocopy knockouts, the efficiency of the AAVMYO-CRISPR/Cas9 method is clearly much higher.

Reviewer #2 (Remarks to the Author):

Line 70-72: The wording of the sentence leaves unclear the fact that the Cre is expressed from a muscle specific promoter (HSA). I would rewrite the sentence to include that information. As it stands it could be misinterpreted that only the tamoxifen induced Cre is muscle-specific.

We rephrased the description of the mice (see lines 71 to 74; highlighted)

Line 173: The figure being referenced is S7A, not S6A.

As we added many data, the numbering has changed. However, we carefully reviewed the new numbering.

Line 343: It's not clear to me how the authors came to the 75% number. If 60% editing in vitro led to >90% knockdown, and the knockdown here is 80% (suggesting less editing), why do the authors argue that real editing must be 75%?

We thank the reviewer for realizing this mistake. We have now reformulated and re-calculated the number (see lines 404 – 405; highlighted).

Line 385: The authors should more explicitly acknowledge that they struggled to balance high levels of local knockdown with systemic leakage. While they could lower the dose of AAVMyo to a point with minimal leakage, local knockdown efficiencies were likewise reduced. For certain genes, this may not be limiting, but for others it could be. While AAVMyo has obvious advantages over AAV9 for systemic delivery, AAV9 may actually be better for individual limb or muscle knockdown experiments based on the authors' data, which showed AAV9 had significantly less transduction due to leak. Additionally, it's possible that local AAV administration could possibly be coupled with local tamoxifen administration in the iCas9mKI mice. Future studies should further refine this method.

Again; thanks for the great suggestion. We have now tested the idea of using the combination of AAV9 and neuraminidase to locally perturb MuSK in the injected muscle only. Indeed, even at the high dose, there was no leakage into muscles of the contralateral leg but high local perturbation in the injected muscle. These data are presented in the new Fig. S12 and described in the result section on page 15; lines 305 – 317 (highlighted). Discussion of these results is on page 22; lines 466 – 470.

Figure S3C: I am curious as to the reason for the reduced Cas9 expression in the C2C12s that received a construct containing the targeting guide RNAs (vs sgNT). If the authors have a theory as to why, it would be nice to have an explanation in the discussion.

Thanks for the question. We really do not know. The most likely explanation is a difference in the transfection efficiency with the expression plasmid coding for sgNT compared to the plasmids containing sgPKC α -1 or sgPKC α -2. Fig. S3 shows the results of 3 independent transfection experiments. The plasmids of each construct derive, however, from a single sample preparation and sgPKC α -1 or sgPKC α -2 expression vectors went through additional cloning steps compared to sgNT to replace sgNT with sgPKC α -1 or sgPKC α -2. Hence, we could have difference in DNA quality of the expression plasmids, which could affect transfection efficiency. Of course, we cannot exclude any alternative explanations.

Reviewer #3 (Remarks to the Author):

The authors highlight that muscle-specific editing can be achieved in Cas9 and Cre-expressing mice. However, this study lacks supporting data to substantiate the specificity of the system.

The authors should prove evidence that systemic or intramuscular delivery of AAV-sgRNA did not trigger genome editing in the heart or liver, as these organs do not express Cas9 protein in this particular mice. In Figure 2, the liver exhibited higher levels of AAV genome compared to TA or heart following intramuscular delivery of AAV. However, the DNA editing was only analyzed in TA muscles, and a comparison of gene editing among these three muscles was not conducted. In Figure 4, as well as in other data, similar results were observed exclusively in skeletal muscles without any comparative analysis with the heart or liver.

We now also include editing at the target site of sgPKC α -1 using TIDE analysis in heart and liver upon **local injection** of AAV9/neuraminidase (new Fig. 2D). No significant editing was observed both in heart and liver despite to high number of AAV genomes/nucleus in liver.

We also measured AAV genomes/nucleus and DNA editing at the target site for sgPKC α -1 after **systemic** delivery of the sgRNA *via* AAV9, AAVMYO, AAVMYO2 and AAVMYO3 (new Fig. S8B, C). While there is some low-level editing in the heart (new Fig. S8C), we did not detect any editing in the liver (new Fig. S8C) despite the tremendously high number of AAV genomes/nucleus when using AAV9 and AAVMYO (new Fig. S8B). Editing in the heart at low level is expected as HSA-Cre is expressed at low level in developing heart (Schwander et al., 2003). We discuss this in the revised manuscript (page 20; lines 408- 420).

In this study, the authors propose that crossing the Cre-dependent Rosa26 with mice that express Cre resulted in muscle-specific Cas9 expression, while the heart and liver did not show such expression. The authors clearly demonstrate how muscle-specific expression is induced in these mice.

In Figure 2A, the authors should provide evidence to support that using three copies of sgPKC α -1 leads to improved genome editing efficiency compared to a single sgRNA treatment group.

We conducted the suggested experiment and show the results in a new figure (new Fig. S4). Indeed, three copies of the sgRNA are more efficient (new Fig. S4C).

In line 106, the authors mentioned that co-injection of neuraminidase with AAV resulted in improved AAV transduction in skeletal muscle. However, the authors should provide evidence by comparing AAV transduction with and without neuraminidase to support this conclusion.

This experiment has been conducted by others (Zhu et al., 2018) and we cite this paper now. Moreover, AAV9/neuraminidase-induced editing at the sgPKC- α 1 target site results in 20.3% (Fig. 2D) and 26.1% (Fig. S4) whereas AAV9 alone results in 17.4% (Fig. 3E). Similarly, the number of AAV genomes/nucleus in presence of neuraminidase in TA muscle is 93 (Fig. 2C), whereas it drops to 34 without neuraminidase (Fig. S5B).

In Figure 6 and 7, it is unclear what the indel frequency is for each of the sgMUSk-1 to 7 and sgACvr2a/b. Further clarification on the genome editing efficiency for each specific sgRNA is needed. In addition, did the use of the sgRNA cocktail result in large gene deletions instead of small indels? The authors should provide data on the mutation patterns to demonstrate the effects of the sgRNA cocktail on genome modifications.

We now measured sgRNA efficiency by amplicon next generation sequencing and CRISPResso2 analysis for each sgRNA targeting *Musk*, *Acvr2a* and *Acvr2b*. The numbers are given in the new figures (new Fig. S10 for *Musk*; new Fig. S13 for *Acvr2a* and *Acvr2b*). Efficiency for each site varies between 6 to 40%.

To characterize large deletions in the genome by sgRNA cocktail, we conducted reverse transcription from polyA-positive mRNA isolated from TA muscle of control and AAVMYO-7sgMusk-injected Cas9mKI mice. PCR amplification of the resulting cDNA using primers that flanked the sites targeted by the 7 different sgRNAs and subsequent sequencing shows a dose-dependent increase in fragmented *Musk* cDNAs, suggesting large genomic deletion in the *Musk* gene (new Fig. S11).

In Figure 5, it is not clear what the relative Musk expression and genome editing frequency in other muscles, such as GAS, EDL, SOL, TRI and MAS.

We now added *Musk* mRNA expression in *gastrocnemius* (GAS), *extensor digitorum longus* (EDL), *soleus* (SOL), *triceps* (TRI) and *masseter* (MAS) muscles (Fig. 5F). *Musk* expression was always below 10%.

In Figure S8, what is the reason behind the increased *Musk* expression in the group treated with AAVMYO-7sgMusk (1x1011vg)?

We are not sure what you are referring to. The only data point where *Musk* expression is increased is in the contralateral TA muscle at 3.3×10^{10} vg (Fig. S8C). While we cannot be certain of the reason, one possibility is that the lowering of MuSK protein in some muscle fibers may cause partial denervation. Denervation is known to cause a strong increase in *Musk* mRNA (Valenzuela et al., 1995) and hence, if there is only partial editing of the *Musk* gene in mice injected with a low dose of AAVMYO-7sgMusk, this upregulation is not prevented.

In Figure 6E, it is observed that the level of postsynaptic AChRs appears to be decreased in the low dose group of AAV-treated mice compared to the high-dose group. The authors should provide quantitative data on the AChR level.

We now show stainings for AChR and MuSK in *gastrocnemius* (GAS) cross sections (new Fig. 6F). We used such cross sections to quantify MuSK (new Fig. 6G) and AChR staining (new Fig. 6H). The density for both proteins was significantly lower in AAVMYO-7sgMusk-injected muscles. Density tended to be lower with high doses of AAVMYO-7sgMusk but these did not become significant. Please also note that mice treated at the highest dose of AAVMYO-7sgMusk needed to be euthanized after only 3 weeks, which may explain the somewhat higher AChR density compared to the lower doses (euthanized only after 5 weeks).

Further clarification is needed to understand the lack of correlation between low expression of *Chma1* etc and the dose of AAVMYO-7sgMusk.

The dosing of AAVMYO-7sgMusk was actually not intended to create a differential response in the injected muscles but to prevent the spreading of the virus to other muscles. Hence, we interpret the results such that all doses used are still sufficient to cause the loss of MuSK at the NMJ of the injected muscles (see cross sections in Fig. 6F and whole mounts in Fig. S9E), which in turn results in denervation and hence upregulation of *Chrna1*, *Chrn3* and *Gadd45a* expression (Fig. 6I-K).

The authors hypothesize that local injection of AAVMYO in the TA muscle may affect other muscles due to systemic circulation of AAV. To confirm this, the authors should quantify the viral genome in the blood after intramuscular injection of AAV, in order to provide evidence for their speculation.

We now present data on AAV transduction of the contralateral TA muscle by measuring the number of AAV genomes/nucleus (new Fig. S9B). At the highest AAVMYO-7sgMusk dose of 3.0×10^{11} vg, the AAV genome number is increased and at this dose, *Musk* expression is down to 10% in the contralateral TA muscle (new Fig. S9C) and the diaphragm (new Fig. S9D). We also report on AAV transduction of heart and liver upon intramuscular injection of AAVMYO-3sgPKC α -1 (new Fig. S5B). The measured values are rather high and the only way to transduce liver is *via* the blood. This route is also used by AAV9 (see AAV genomes/nucleus in Fig. S5B).

The authors should provide the mismatched sequence information at the potential off-target sites. Additionally, it is important to analyze the off-target effects following systemic delivery, particularly in the highest dose of AAVMYO, to comprehensively assess the potential off-target effects of the treatment.

We now provide the editing data for the primary off-targets of all sgRNAs as predicted by CRISPOR in new Table S4 and primary off-target editing efficiency (new Fig. S10C, new Fig. S13D-E, new Tables S2 and S3), evaluated by amplicon next generation sequencing and CRISPResso2 analysis. There are two off-target sites that show increased editing but they reside in an intergenic or an intronic region, respectively, which makes it unlikely to result in a major off-target effect.

The authors should perform a statistical comparison of the data between AAV9 and AAVMYO (Fig 3D-G, Fig 4F-G, Fig S5A), as well as between different disease dose (Fig 6D; 1x1011vg vs 3x1011vg), to thoroughly assess any differences and provide a comprehensive analysis.

Statistical differences are significant between AAV9 and AAVMYO and they are indicated on top left of each graphs (e.g. 9vsMYO**) in Fig. 3D-G, Fig. 4F-G and Fig. S5A. We applied statistics for differences between doses in Fig. 6D (now Fig. 6C), but no significance was reached.

Please provide a statistical analysis to specify the difference observed in Figure S9B.

Statistical analysis was done but the difference is not significant (now Fig. S14B). The lack of significance was never indicated in any of the figures.

References:

- Schwander, M., Leu, M., Stumm, M., Dorchies, O.M., Rugg, U.T., Schittny, J., and Muller, U. (2003). Beta1 integrins regulate myoblast fusion and sarcomere assembly. *Dev Cell* 4, 673-685.
- Valenzuela, D.M., Stitt, T.N., Distefano, P.S., Rojas, E., Mattsson, K., Compton, D.L., Nunez, L., Park, J.S., Stark, J.L., Gies, D.R., Thomas, S., Lebeau, M.M., Fernald, A.A., Copeland, N.G., Jenkins, N.A., Burden, S.J., Glass, D.J., and Yancopoulos, G.D. (1995). Receptor Tyrosine Kinase Specific for the Skeletal-Muscle Lineage - Expression in Embryonic Muscle, at the Neuromuscular-Junction, and after Injury. *Neuron* 15, 573-584.
- Zhu, H., Wang, T., John Lye, R., French, B.A., and Annex, B.H. (2018). Neuraminidase-mediated desialylation augments AAV9-mediated gene expression in skeletal muscle. *J Gene Med* 20, e3049.

Reviewers' Comments:

Reviewer #1:

Remarks to the Author:

My apologies to the authors for not simply sharing with them the comments I made to the editor regarding this manuscript so here they are:

"My problem with Methods papers is that they too frequently seem to report clever and cool experimental methods that promise to provide new insight into biology, but for whatever reason the authors stop short of using their own method to discover something new. Similarly, this study falls short of interesting (for me) because the authors didn't attempt to employ their new method to assess the functional relevance of just one new gene in skeletal muscle. Perhaps they didn't go any further because they only achieved knockdown (and not knockout as claimed a few times throughout) in 4 out of 4 genes targeted (0% knockouts), which leaves open the high probability that any non-phenotypic knock downs resulting from this method will only delay (and increase the cost) of experiments to assess the relevance of a gene in skeletal muscle via more classic knockout approaches. Thus, the promise of this somatic gene deletion method reminds me of the Cologuard colon cancer screening ads: Cologuard may alert you that you could have colon cancer, but you'll still need a colonoscopy to know for sure, or worse, it may falsely report that you don't have colon cancer."

Reviewer #2:

Remarks to the Author:

The authors have developed a clever way to knock down genes using AAV to deliver guide RNAs to muscle following systemic or local injections in transgenic mice expressing constitutive or inducible Cas9 in skeletal muscle. This model would serve as an alternative to knockdown strategies using RNAi, which are harder to tissue restrict and can be lost over time. I believe that their established mouse models and systemic and local knockdown protocols described here will be heavily utilized by the field in the future.

The edits in response to reviewer comments have addressed the major shortcomings of the original submission. The data analysis is appropriate, and their conclusions are supported by the data.

I recommend that this article be published.

Signed,

Julie M. Crudele

Reviewer #3:

Remarks to the Author:

The authors addressed all issues raised by the reviewers.